# Double-strand break repair pathways differentially affect processing and transduction by dual AAV vectors

Anna C. Maurer [1,2] ✉, Brian Benyamini [1], Oscar N. Whitney[1], Vinson B. Fan [1], Claudia Cattoglio[1,2,3], Djem U. Kissiov [1,3], Gina M. Dailey [1,4], Xavier Darzacq[1,4], Matthew D. Weitzman[5] & Robert Tjian [1,3,4]

Recombinant adeno-associated viral vectors (rAAV) are a powerful tool for gene delivery but have a limited DNA carrying capacity. Efforts to expand this genetic payload have focused on engineering the vector components, such as dual trans-splicing vectors which double the delivery size by exploiting the natural concatenation of rAAV genomes in host nuclei. We hypothesized that inefficient dual vector transduction could be improved by modulating host factors which affect concatenation. Since factors mediating concatenation are not well defined, we performed a genome-wide screen to identify host cell regulators. We discover that Homologous Recombination (HR) is inhibitory to dual vector transduction. We demonstrate that depletion or inhibition of HR factors BRCA1 and Rad51 significantly increase reconstitution of a large split transgene by increasing both concatenation and expression from rAAVs. Our results define roles for DNA damage repair in rAAV transduction and highlight the potential for pharmacological intervention to increase genetic payload of rAAV vectors.

Recombinant adeno-associated viral vectors (rAAV) are a leading platform for clinical DNA delivery. This vector system is derived from a human parvovirus and has many inherent features that make it attractive for human gene therapy applications[1]. It also has some inherent limitations, the major one of which is the small genetic payload. The unenveloped 25 nm capsid can package no more than 5 kb of single-stranded DNA (ssDNA) into its small lumen[2,3], imposing a size limit to the coding and regulatory elements that can be included in the transgene cassette. For gene replacement therapies, current strategies employ cDNA versions of the gene of interest, with expression driven by a short, strong promoter such as CMV[4]. While this accommodates many therapeutically relevant gene sizes, it excludes other larger genes, such as dystrophin or the most accurate Cas9 versions and other base editors. Moreover, CMV and similar promoters are ubiquitously expressed at very high levels, leading to off-target expression in non-therapeutically relevant tissues, and potentially toxic levels of transgene expression. This can lead to immune responses and other failures of the therapy. A larger payload size is therefore highly desirable for expressing large proteins and moreover for enabling fine-tuned spatiotemporal control over expression and levels by inclusion of more diverse, efficient regulatory sequences.

The wild-type (wt) AAV virus and the recombinant AAV vector have the same overall structure, consisting only of the 60mer icosahedral capsid with one packaged ssDNA vector genome (VG). Viral proteins are required during production for replication and packaging, but no viral enzymes are packaged within the virion[5]. VG architecture requires flanking the desired delivery DNA with the viral inverted terminal repeat (ITR) sequences at both ends[6]. The ITR contains

[1]Department of Molecular and Cell Biology, University of California, Berkeley, CA, USA. [2]CIRM Center of Excellence, University of California, Berkeley, CA, USA. [3]Howard Hughes Medical Institute, University of California, Berkeley, CA, USA. [4]Li Ka Shing Center for Biomedical & Health Sciences, University of California, Berkeley, CA, USA. [5]Department of Pathology and Laboratory Medicine, University of Pennsylvania Perelman School of Medicine and the Children's Hospital of Philadelphia, Philadelphia, PA, USA. ✉e-mail: acmaurer@umich.edu

sequence elements that are required during viral replication and also direct packaging into the preformed capsids for vector production[7]. In the transduction process (Fig. 1A), the capsid traffics the VG to the host/target cell nucleus, where the capsid is shed (termed uncoating)[8,9]. The ssDNA VG is converted to dsDNA through unclear mechanisms which enables transgene expression. Individual VGs can also circularize through intramolecular ITR-ITR joining and concatenate through intermolecular ITR-ITR joining[10]. In its recombinant form, where a capsid of choice is packaged with an ITR-flanked transgene, rAAV VGs cannot replicate; but instead multiple VGs enter the nucleus and can concatenate into large episomal species[11]. The absence of delivered enzymes implies that the post-uncoating nuclear steps of rAAV transduction are orchestrated by host cellular factors. Decades of observation have articulated the nuclear steps of transduction—uncoating, conversion to dsDNA, concatenation, expression —yet they remain observations, with little mechanistic detail and knowledge of facilitating host factors[12,13].

The absence of viral enzymes in the rAAV particle implies that the processes of VG circularization and concatenation are orchestrated by host cellular factors. ITR-ITR joining is analogous to host DNA repair mechanisms, and dsDNA VGs could conceivably resemble a chromosomal break. The host cell has many ways of repairing DNA damage, including two main pathways to repair double-strand breaks (DSB): the high-fidelity but restricted activity pathway called Homologous Recombination (HR) or Homology Directed Repair (HDR), and the constitutively active but error-prone pathway of Non-Homologous End Joining (NHEJ)[14]. The MRE11/RAD50/NBS1 (MRN) complex, which recognizes free DNA ends to facilitate repair upstream of HDR versus NHEJ pathway choice, recognizes ITRs but inhibits concatenation[15,16]. The NHEJ pathway has been suggested to play a role in concatenation since depletion or inhibition of the NHEJ factor DNA-PKcs reduces VG concatenation[17–21]. However, the effect of HDR pathway on concatenation has not been directly examined.

The propensity of VGs to concatenate can be exploited to increase the final genetic payload size[22]. In this "dual vector" approach (Fig. 1A, B), a large transgene cassette is split into two halves, packaged into separate capsids, and co-delivered to target cells. Inclusion of a splice donor in the "Left" half VG (L) and a splice acceptor in the "Right" half VG (R) allows the ITR junction to be spliced out during RNA processing of transcription, and the large gene is subsequently reconstituted in the transcript. This technique was first developed in the early 2000s by multiple groups independently[23–25]. To be effective it

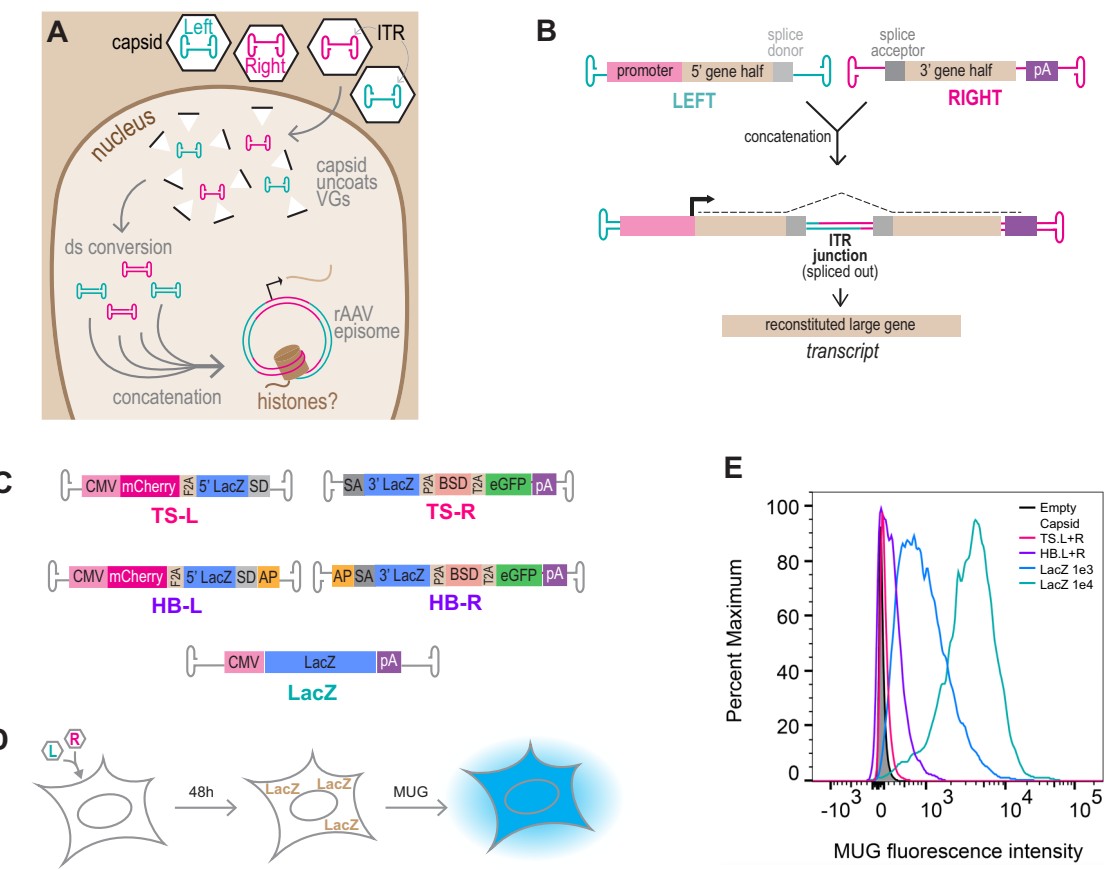

**Fig. 1 | Dual Vectors are inefficient. A** Schematic of recombinant adeno-associated viral (rAAV) vector genome (VG) processing. Single-stranded DNA VGs are converted to double-stranded (ds)DNA and VGs circularize and concatenate by inverted terminal repeat (ITR)-ITR joining, through unclear mechanisms. VGs associate with histones, but when this occurs during these processes is unknown. **B** The dual vector approach can double payload size by exploiting the natural concatenation process. A large gene is split into separate Left (L) and Right (R) VGs; inclusion of a splice donor (SD)/acceptor (SA) excludes the ITR junction from the final transcript. **C** Reporter constructs to assay split transgene reconstitution. LacZ is split between L and R VGs in a trans-splicing (TS) pair that requires ITR-ITR concatenation for expression, or a Hybrid (HB) trans-splicing plus Alkaline Phosphatase (AP) partial sequence previously demonstrated to increase split transgene

reconstitution efficiency. An unsplit LacZ VG is used as a control. CMV = cytomegalovirus. BSD = Blasticidin resistance. pA = poly A signal. F2A, P2A, T2A = 2A self-cleaving peptides. **D** LacZ Activated Fluorescence Assay (LAFA) schematic: cultured cells are transduced with equimolar doses of L and R vectors. After 48 h, split transgene reconstitution is assayed by Beta-galactosidase (LacZ) activity cleaving 4-Methylumbelliferyl-beta-D-galactopyranoside (MUG), a fluorogenic beta galactose analog. **E** U2-OS cells were treated with 1e4 VG/cell of the split pairs or unsplit LacZ diagrammed in (**C**). LAFA was performed at 48 hours post transduction and quantified by flow cytometry: L + R pairs (magenta, purple), unsplit vector (blue, green), or empty capsid (black) negative control are plotted as a histogram. All capsids and ITRs are AAV2.

requires correct, directional, and stoichiometric concatenation, followed by correct expression. By nature of the sequence, any ITR can in theory join any ITR, leading to many permutations that do not all lead to correct split transgene reconstitution. The approach also currently relies on very high vector doses. To increase the chances of L and R genomes joining in the correct orientation, a short fragment of the Alkaline Phosphatase (AP) gene serendipitously found to be "highly recombinogenic" in the dual vector context[26] can be added between the splice signal and the ITR[27]. It is presumed that the AP fragment promotes annealing of + and – sense VGs which then become a reconstituted large gene through HR. However, HR has never been directly tested as the mechanism of concatenation. Indeed, AP inclusion significantly increases efficiency and is now commonly used in dual vectors, but like general mechanisms of concatenation, little is known about the mechanisms underpinning the AP phenomenon.

Here, we aim to improve dual vector transduction by an alternative approach that focuses on the host cell. We first identify cellular pathways inhibitory to split transgene reconstitution by a genome-wide screen. We provide evidence for early chromatinization of VGs, and epigenetic recognition as a double-strand break. Surprisingly, blocking HR increases both concatenation of and expression from rAAV VGs, which dramatically increases dual vector transduction efficiencies. Our findings suggest that HR is the "first responder" to the DSB-flagged VGs, but that NHEJ is a more efficient mechanism for VG concatenation. While the majority of efforts to improve rAAV gene therapies have focused on modifying the vector and dose, our findings emphasize the importance of understanding host responses in the target cell. Our results highlight the potential for pharmacological approaches to enhance efficiency and increase deliverable transgene size in rAAV gene therapies.

## Results

### A genome-wide screen to identify cellular inhibitors of split transgene reconstitution

To identify cellular factors whose modulation could increase dual vector transduction, we established a platform amenable to pooled genome-wide screens by building upon dual vector systems previously engineered by others in which vectors enable trans-splicing (TS) or contain the additional AP segments in Hybrid (HB) vectors[23–25,27–29] (Fig. 1C). Correct directional concatenation and successful expression will yield a multi-cistronic cassette containing several reporter genes. The LacZ gene is split to span the ITR junction between Left (L) and right (R) vectors and is therefore the most stringent reporter of concatenation and subsequent expression. After transducing cultured cells, incubation with the fluorogenic B-galactose analog 4-Methylumbelliferyl-beta-D-galactopyranoside (MUG) provides an enzymatic readout of successful split transgene reconstitution. This LacZ activated fluorescence assay (LAFA) can be performed on cell lysates in microwell plates or in flow cytometry based approaches and enables FACS isolation of transduced cells (Fig. 1D). We transduced U2-OS cells with unsplit LacZ as a positive control, with either the TS or HB pair of dual vectors, or an empty capsid as a negative control, and used LAFA and flow cytometry to assay expression of LacZ. Flow cytometry of treated cells (Fig. 1E) is consistent with previous observations by others[29]: although HB dual vectors (purple) achieve higher transduction than TS (magenta) at a moderate dose of 1e4 total vg/cell (5e3 VG/cell of L vector plus 5e3 VG/cell of R vector), both the TS and HB dual vectors transduced at efficiencies two orders of magnitude lower than cells receiving the same dose of an unsplit LacZ vector (green). The inefficiency of dual vector transduction is further demonstrated by comparing 1e4 total VG/cell dual vectors (purple) to 1e3 VG/cell unsplit LacZ (blue); a 10-fold lower dose of unsplit transgene achieves 10-fold higher transduction than dual vectors. To ensure that LacZ activity measured after dual vector transduction is from VG-VG recombination and not transcripts recombining, we transfected plasmids containing the split vector genomes and unsplit controls (Supplementary Figs. 1

A–F), which do not have free ITRs and thus don't concatenate as capsid-delivered VGs do, but actively express from their promoters. We do not observe LacZ reconstitution in this setting (Supplementary Fig. 1G), suggesting that concatenation must occur at the DNA level for a spliced transcript to reconstitute a split gene.

To identify opportunities for host factor modulation that could lead to increases in split transgene reconstitution, we reasoned that (1) most pharmacological intervention presents inhibitors rather than activators, and (2) as a virally derived entity, it is likely that rAAV concatenation and expression are targets of negative regulation as part of an innate defense. We therefore designed a genome-wide knockout (KO) screen to identify repressors of dual vector transduction. Lentiviral Brunello libraries carrying a puromycin resistance cassette, Cas9, and an average of four guides per human gene were used to generate genome-wide KO libraries of U2-OS cells. After Puro selection, the cellular KO libraries were co-transduced with 5e3 VG/cell of each of the HB-L and HB-R vectors. Cells were collected at 48 hours post-transduction (hpt), stained with MUG, and LacZ expressing cells isolated by FACS. Genetic perturbations in these cells were identified by Next Generation Sequencing (NGS), and statistical analysis on guide overrepresentation was performed to generate a list of candidate inhibitors of dual vector transduction (Fig. 2A). Pathway analysis of the top 5% of significantly enriched perturbations in the MUG-positive pool (Supplementary Data 1) revealed overrepresentation in DNA synthesis and repair, particularly double-strand break (DSB) repair. Of the two major DSB repair pathways, Homology Directed Repair (HDR) and Non-Homologous End Joining (NHEJ), only HDR was overrepresented (Fig. 2B). Since the NHEJ pathway has been suggested to positively regulate concatenation, we hypothesized that the HDR DSB repair pathway could function as a possible negative regulator of VG concatenation. We explored this in our subsequent hit validation.

### Vector genomes are epigenetically marked as double-strand breaks

Since chromosomal DSBs are marked by phosphorylation of the histone variant H2AX (γH2AX) which coordinates recruitment of repair factors, we examined whether VGs are subject to this epigenetic mark. Previous studies have observed an increase in nuclear γH2AX when cells are infected with wt AAV[30–32], but this epigenetic mark has not yet been examined on AAV viral or vector genomes themselves. To detect individual VGs, we adapted a previously established approach[15,33–36] in which the rAAV genome is visually detected through recognition of lacO binding sites by a fluorescent LacI fusion protein. We generated U2-OS cells stably expressing the LacI-mNeonGreen fusion protein; untransduced cells have no LacI-mNG foci (Supplementary Fig. 2A) and normal γH2AX immunostaining patterns, with foci distributed throughout nuclei (Supplementary Fig. 2B). Etoposide was used as a control for γH2AX staining under DNA damage conditions. We transduced these cells with rAAV vectors containing VGs consisting of 64 LacO repeats. After uncoating and conversion to a dsDNA species, the VG recruits many fluorophores and can be visualized in live or fixed cells (Fig. 3A). We observed VG foci distributed throughout the nucleus, fusing with other foci, and increasing in size and intensity over time, indicating concatenation events (Supplementary Movies 1-6). We first asked whether VG foci are marked as DSBs by immunostaining for γH2AX and observed colocalization as early as 4 hpt (Fig. 3B). High-content imaging analysis on 1139 VG foci at 4 hpt revealed 64.97% colocalization with γH2AX foci. To verify γH2AX association with VGs, and nucleosome incorporation in general, we performed ChIP-qPCR for γH2AX and H2B, respectively (Fig. 3C). At the earliest timepoint for which ChIP signal was detectable above background (12 hpt), both γH2AX and H2B precipitated VGs, corroborating our imaging observations. To date, this is the earliest observed histone association with rAAV-delivered DNA and suggests that VGs are both chromatinized and flagged as DSBs quickly after uncoating.

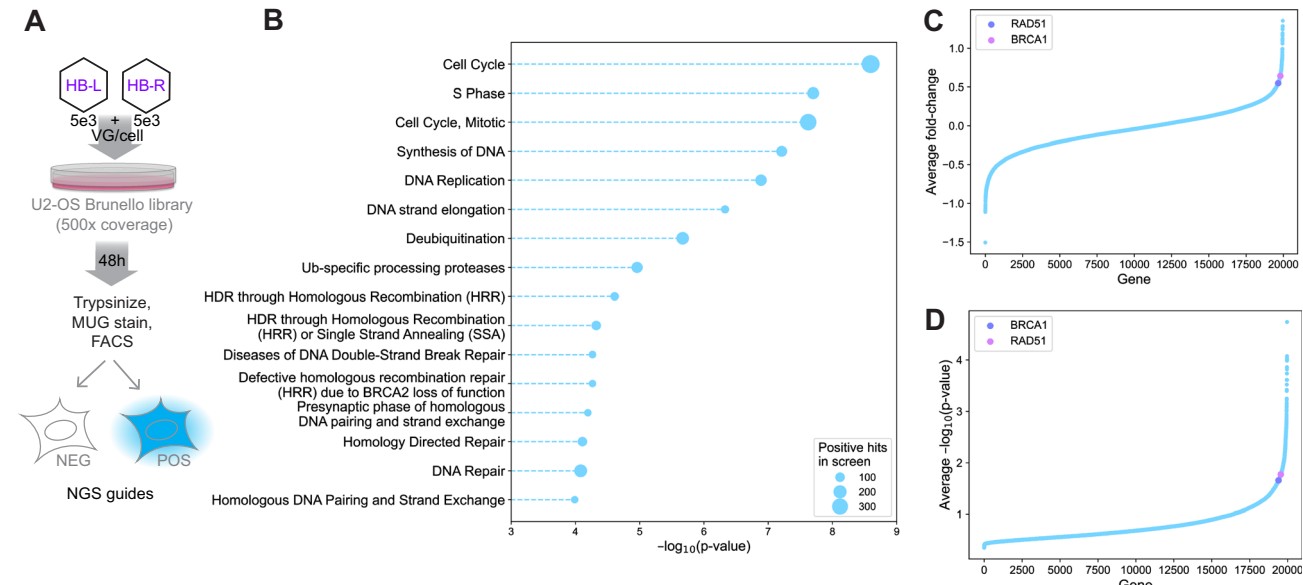

**Fig. 2 | A genome-wide screen to identify cellular inhibitors of split transgene reconstitution. A** Screen schematic: a Brunello knock-out Library of U2-OS cells (500x coverage) was treated with 5e3 VG/cell of the HB dual vectors. Cells were collected at 48 hours post transduction, stained with 4-Methylumbelliferyl-beta-D-galactopyranoside (MUG), and MUG negative (NEG) and positive (POS) cells separated by FACS. **B** Reactome pathway analysis results on the top 5% of genes in the positive population (candidate negative regulators of split transgene reconstitution), sorted by significance. Bubble size indicates the number of genes in each Reactome category. HDR = homology directed repair. **C** Guide-enrichment score (sgRNAs enriched in the MUG-positive population) sorted in ascending order and plotted as a waterfall plot. Rad51 and BRCA1 are highlighted. **D** -log10(*p*-value) of guide enrichment scores sorted and plotted in ascending order. Rad51 and BRCA1 are highlighted. See Methods for analysis details and plotting code.

We next used immunostaining to ask whether VGs recruit DSB repair factors. We used antibodies to BRCA1 as a representative HR factor which was enriched in our screen, and antibodies to 53BP1 as a representative NHEJ factor (Fig. 3D). Colocalization of both repair factors were observed with VG foci, with BRCA1 more often than 53BP1. We quantified overlap of these foci in 957 nuclei (Fig. 3E); BRCA1 colocalizes with 72.4% of LacI mNG foci, whereas 53BP1 is 43.2%. 36.9% of all VG foci colocalize with BRCA1 but not with 53BP1, whereas 53BP1 colocalizes without BRCA1 on only 7.4% of VGs. This suggests that DSB repair factors are recruited to most newly uncoated VGs, and that HDR machinery may be preferentially recruited over NHEJ. These anti-correlated staining patterns are in line with our opposing screen results for NHEJ and HDR, and the known competition between these DNA repair pathways on DSBs in the cellular genome[14].

## HDR deficiency increases concatenation and expression from rAAV vectors

Successful trans-splicing dual vector transduction requires two major steps: concatenation of the L and R co-infected VGs followed by successful expression of the split cassette. To examine whether our candidate factors affect concatenation or expression, we added an mScarlet expression cassette downstream of the lacO array (lacO.64.CMV.mScarlet; Fig. 4A). This enables simultaneous and quantitative visual readouts of both concatenation (mNeonGreen foci) and expression (mScarlet intensity) which are not interdependent, such as with dual vector transduction and LacZ reconstitution. To test effects of HDR loss on concatenation and expression, we introduced BRCA1 or scrambled guides by nucleofection into U2-OS^LacI-mNG,Cas9 cells, waited 72 h for editing, and then transduced with rAAV2/2.lacO.64.CMV.mScarlet vectors. Immunostaining for BRCA1 showed a significant decrease in cells receiving BRCA1 guides compared to scrambled guides (Supplementary Fig. 3A). We observed a 2.3-fold increase in the average number of VG foci per cell (Supplementary Fig. 3B) and a significant increase in the percentage of cells with any VG foci in cells receiving BRCA1 guides compared to Scramble

(Supplementary Fig. 3C). We next quantified expression by mScarlet intensity in the high-content images. Expression of the mScarlet transgene is 10% higher in cells receiving BRCA1 guides, a small but statistically significant increase (Supplementary Fig. 3D). Together these observations suggest that BRCA1 inhibits concatenation and de novo episome formation, with a small impact on expression.

To orthogonally validate this result, we next depleted BRCA1 with siRNA and observed nearly complete knockdown by 26 h after transfection (Fig. 4E). We transduced BRCA1 knockdown cells with 1e5 VG/cell of rAAV2/2.lacO.64.CMV.mScarlet at this timepoint and observed significant increases in LacI foci per nucleus (Fig. 4B), percentage of cells with any VG foci (Fig. 4C), and mScarlet expression (Fig. 4D). This exceeded the increases observed in the CRISPR/Cas9 acute KO. In parallel, to interrogate whether the increased foci formation is specific to BRCA1 loss or is an effect of non-functional HDR, we tested the effects of Rad51 siRNA knockdown, another HDR factor enriched in our screen. At 26 h, the knockdown was not complete, but Rad51 levels did not deplete further upon longer incubation (Fig. 4E). Cells were therefore transduced at 26 h post-siRNA transfection (Fig. 4B–D) and we observed similar increases to those observed with BRCA1 loss across all parameters. As an orthogonal method of examining Rad51 loss in transgene expression and concatenation and to test for dose-dependent effects, we inhibited Rad51 with the drug B02 by pretreating U2-OS^LacI-mNG cells with multiple doses of drug, then transduced with multiple doses of rAAV2/2.lacO.64.CMV.mScarlet. We observed that B02 increased the number of VG foci per cell (Fig. 4F, G) and the percentage of cells forming any VG foci (Fig. 4H), in a dose-dependent fashion. In live imaging of VG foci, we observe more fusion events in B02 treated cells (Supplementary Movies 1-3) than DMSO treated cells (Supplementary Movies 4-6). Additionally, high-dose Rad51 inhibitor resulted in approximately doubled mScarlet expression, whether vector was at moderate (1e4 VG/cell) or high doses (1e5 VG/cell) (Fig. 4I). B02 does not affect Rad51 protein levels (Fig. 4E), suggesting that it is the ATPase *activity* of Rad51 contributing to the inhibitory effects on rAAV. These results suggest that, similar to BRCA1, Rad51

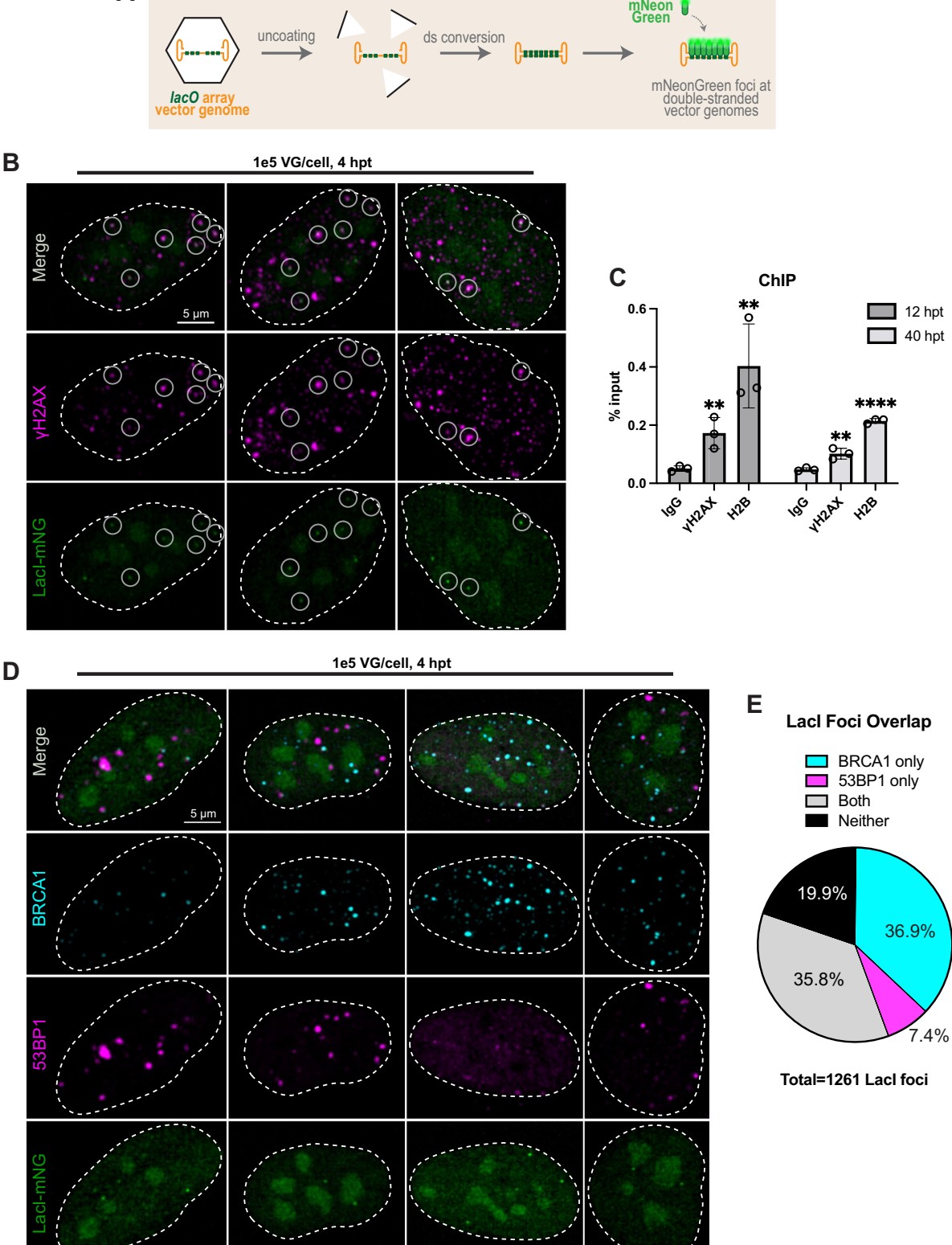

inhibits concatenation, de novo episome formation, and transgene expression. Together these results implicate repressive roles for HDR in rAAV transduction.

To examine whether the observed increases in VG foci and transgene expression were due to increased total number of VGs present in cells, for example due to increased nuclear import or increased dsDNA conversion leading to increased stability/decreased degradation in HDR deficient cells rather than increased concatenation or expression activity, we compared VG copy number in DMSO and B02 treated cells 48 hpt. Cells treated with B02 have no significant difference in VG copy number compared to cells treated with DMSO and receiving the same vector dose (Supplementary Fig. 3E). Taken together, these observations confirm that HDR inhibits both concatenation and expression of rAAV VGs.

**Fig. 3 | rAAV vector genomes are recognized as double strand breaks. A** Vector genome (VG) visualization schematic. Capsids are packaged with array VGs comprised of 64 lacO repeats. After uncoating and becoming double-stranded (ds) in LacI-mNeonGreen expressing cells, VGs can be visualized as green foci. U2-OS[LacI-mNeonGreen] cells were transduced with 1e5 VG/cell of AAV2/2.LacO.64 in all panels. **B** Cells were fixed at 4 hours post transduction (hpt) and immunostained for γH2AX. Images are representative of biological triplicate experiments. **C** Cells were collected at 12 or 40 hpt and chromatin immunoprecipitation (ChIP) was performed with antibodies on *x*-axis in biological triplicate with individual replicates plotted as open circles. qPCR to quantify VGs is reported as a percentage of input material; bars represent mean and error bars = SD. IgG versus histone conditions compared by unpaired, two-tailed *t*-test, ** $p < 0.01$; **** $p < 0.0001$ (**D**) Cells were fixed at 4 hpt and immunostained for BRCA1 and 53BP1, then (**E**) foci overlap quantified in 957 nuclei. Images and quantification are representative of biological duplicate experiment. Source data for plots are provided as a Source Data file.

Because DDR inhibition can affect the cell cycle, we examined these effects by treating cells with drug or siRNA and examining DNA content 48 h later by flow cytometry (Supplementary Fig. 4). BRCA1-siRNA does not significantly affect the proportion of cells in G1, S, or G2/M compared to the negative control siRNA (maximum of 3.5% difference, Supplementary Figs. 4F, H), whereas RAD51 siRNA has a significant effect (maximum 9.9% difference, Supplementary Figs. 4F, G). Comparing B02 to DMSO conditions, however, demonstrates pharmacological Rad51 inhibition does not recapitulate the effect on cell cycle of siRNA (Supplementary Figs. 4A, B). Since we observe the strongest increases in VG foci formation under BRCA1 depletion and this perturbation has the smallest effect on cell cycle, it is unlikely that cell cycle differences induced by HDR loss are indirectly driving the observed transduction effects, although we have not completely ruled this out. Because HR factors are upregulated during S-phase, we also examined whether BRCA1 preferentially colocalized with VG in S-phase by EdU labeling and immunostaining (Supplementary Fig. 4 C-H). Comparing cells in S-phase to those outside of S-phase, there were no obvious differences in BRCA1/VG colocalization, as well as VG foci per cell and percentages of cells with any foci.

## ATM and ATR inhibition increase transgene expression

The histone variant H2AX can be phosphorylated by ATM upon DSB detection, or by ATR upon replication stress. To examine this, we pharmacologically inhibited these kinases (Supplementary Fig. 3 F-J). We observe a decrease in γH2AX/VG colocalization under ATM inhibition, whereas ATR inhibition has no effect (Supplementary Fig. 3F). This suggests that ATM phosphorylates H2AX associated with VGs. It should be noted that ATM inhibition decreases global γH2AX staining and ATR inhibition does not (Supplementary Fig. 3G). Consistent with previous reports[37,38], compared to DMSO controls we observed an increase in transgene expression under ATM inhibition (Supplementary Fig. 3H). ATR is activated upon wt AAV2/Adenoviral coinfection[31] and UV-inactivated AAV2 monoinfection[39], and ATR inhibition has been shown to decrease wt AAV DNA replication[31], but ATR is not well-studied in the rAAV transduction setting. We observed a large increase in transgene expression under ATR inhibition (Supplementary Fig. 3H). Conversely, these drugs have little effect on the number of LacI foci per nucleus (Supplementary Fig. 3I), and ATR inhibition, but not ATM inhibition, significantly increases the percentage of cells with any LacI foci (Supplementary Fig. 3J). The large effects of ATR inhibition on transduction may not depend on γH2AX but may be due to proteomic changes that affect HR in these conditions[40].

## Rad51 inhibition increases the proportion of transcriptionally active vector genomes

We next asked whether the increase in expression we observe under Rad51 inhibition is a result of increased transcriptional activity from each active VG or an increase in the proportion of VGs that are transcriptionally active. To observe transcriptional activity in situ with each VG, we replaced the mScarlet gene with an MS2 array which forms stem-loops when transcribed. In MCP-mCherry expressing cells the nascent transcript can be visualized as an mCherry focus overlapping with an mNeonGreen (VG) focus (Fig. 4J). Transducing U2-OS[LacI-mNG, MCP-mCherry] cells with rAAV2/2.lacO.64.CMV.MS2 array

vectors, we observed large VG foci often with multiple actively transcribing units within the concatemer (Fig. 4K). After pretreatment with B02 or DMSO, high-content imaging at 48 hpt revealed a significant increase in the percentage of VG foci with at least one overlapping MS2 focus in B02 treated cells (Fig. 4L). This suggests that Rad51 inhibition increases expression by increasing the proportion of transcriptionally active genomes.

Given the inhibitory role we observed for HDR in concatenation and expression independently, we next tested the effects of B02 on dual vector transduction, which requires both processes. We pretreated cells with multiple doses of B02, transduced cells with multiple doses of the Trans-Splicing (TS) L + R split LacZ vector (Fig. 1C), and used LAFA (Fig. 1D) in a 96-well format to readout split transgene reconstitution at 48 hpt (Fig. 5A). At all dose combinations, B02 significantly increased dual vector expression. The most dramatic effect was seen at the lowest vector doses, at which 10 mM B02 increased split LacZ expression 27-fold over DMSO treated cells receiving the same vector dose. This B02-induced increase in expression at low vector doses is comparable to expression levels achieved by 100-fold higher vector doses not treated with B02 (Fig. 5A, gray dashed line). This suggests that pharmacologically induced improvements in transduction will enable significantly lower doses of vector to be used for gene replacement approaches. These effects are not limited to osteosarcoma cells, as we observed a similar trend in trophoblast cells, albeit to a lower magnitude (Supplementary Fig. 5).

## The AP fragment increases concatenation by an HR-independent pathway

While the TS dual vectors serve to model the natural concatenation process and best represent our high-content imaging findings, our original screen was performed with the more efficient hybrid (HB) dual vectors (Fig. 1C), for reasons of experimental practicality. Originally developed by Ghosh and colleagues[27], this pair contains a "highly recombinogenic" overlapping sequence fragment from the alkaline phosphatase (AP) gene that bridges L and R vectors, providing an additional mode of reconstitution beyond the ITR-ITR concatenation in the traditional TS dual vectors (Fig. 1B & C). The mechanism through which AP increases efficiency of dual vector transduction is not well understood; it is suggested in the literature that the AP overlapping sequence drives split transgene reconstitution through homologous recombination[26,27], however HR or HDR have not yet been directly examined in this context. Surprisingly, B02 had even more potent effects increasing HB dual vector transduction than TS (Fig. 5B), suggesting that the classical HR pathway is not the main mechanism through which hybrid dual vectors reconstitute a split transgene. Rad51 inhibition also increased unsplit LacZ vector transduction (Fig. 5C), but to a much lower degree than dual vectors; at 1e4 VG/cell, unsplit LacZ expression is increased 4-fold by high dose B02, whereas L + R vectors at the same dose are increased up to 34 fold with B02 (Fig. 5A–C). This further corroborates our findings that HDR loss increases expression from VGs, and suggests that the concomitant increase in concatenation synergistically contributes to a dramatic increase in dual vector transduction.

The observation that HDR inhibits VG concatenation is counterintuitive, since concatenation could be considered as analogous to

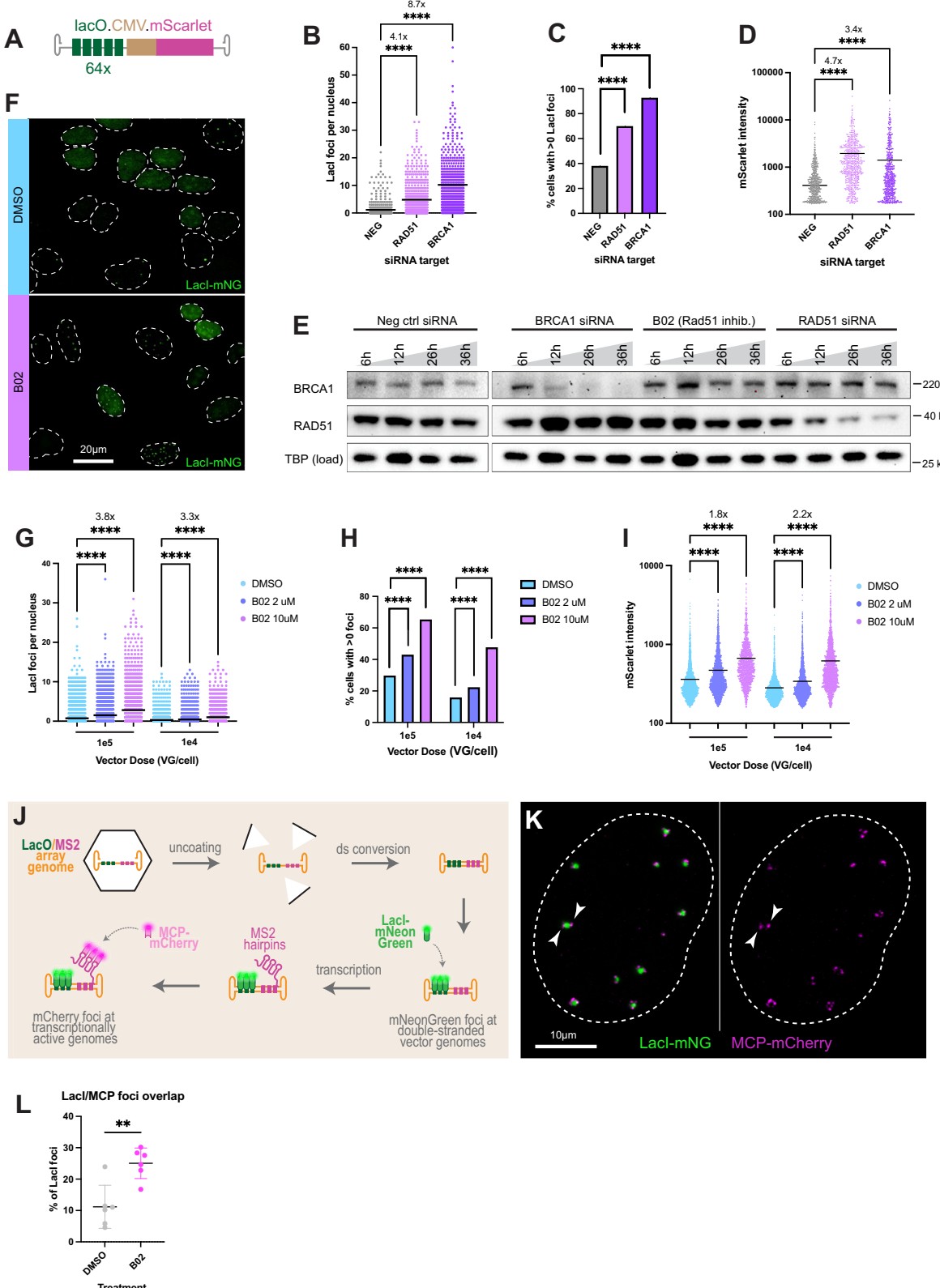

repairing a chromosomal DSB. It is not clear what VG structures are flagged as DSBs or other forms of damaged DNA, but possibilities include the blunt ends after ds-conversion[41] (among other conformations, diagrammed in Fig. 6). HDR is a less active repair mechanism than its error-prone counterpart Non-Homologous End Joining (NHEJ), which coordinates DSB repair through factors distinct from HDR including 53BP1 and the DNA-dependent protein kinase catalytic

subunit (DNA-PKcs). Previous studies recovered fewer circularized VGs from transduced muscle in DNA-PKcs deficient mice than in controls[18,42], suggesting an important role for NHEJ in VG concatenation. We next tested how NHEJ affects split transgene reconstitution by pharmacologically inhibiting DNA-PK with the drug Nu7441 (Fig. 5D–F). We observed a significant reduction in dual vector (Fig. 5D, E) or single vector transduction (Fig. 5F) under DNA-PK

**Fig. 4 | HDR deficiency increases concatenation and expression from rAAVs.**
**A** VG schematic. **B**–**D** U2-OS^LacI-mNeonGreen cells were transfected with siRNA against RAD51 or BRCA1, or nontargeting (NEG) control (*x*-axis), then transduced 26 h later with 1e5 VG/cell AAV2/2.lacO.CMV.mScarlet. High-content imaging was performed at 48 hours post transduction (hpt), and quantitative analysis used to define (**B**) individual cells plotted as points by number of LacI foci per nucleus, and (**C**) the percentage of all imaged cells with any LacI foci. **D** Images were segmented by cell and transgene expression levels assessed by average mScarlet pixel intensity, then plotted as individual points. **B**, **C**, & **D** are plotting 590 cells per condition, and is representative of biological triplicate experiments. **E** Western blot to examine protein levels in cellular lysates 6–36 h after siRNA or drug treatment. Blot is representative of biological triplicate experiments. (**F**–**I**) U2-OS^LacI-mNeonGreen cells were pretreated with drug or dimethyl sulfoxide (DMSO) (legend), transduced with AAV2/2.lacO.CMV.mScarlet at dose on *x*-axis, then high-content imaged at 48 hpt. **F** Primary image example. **G** Individual cells are plotted as points by number of foci per nucleus. **H** Percentage of all cells in (**G**) with any LacI foci. **I** Individual cells are plotted as points according to mScarlet intensity. **G**, **H**, & **I** are plotting 1757 cells per condition, and is representative of biological triplicate experiments. **J** Schematic of lacO.MS2 system to quantify transcriptionally active VGs. **K** U2-OS^LacI-mNeonGreen, MCP-mCherry cells were treated with 1e5 VG/cell AAV2/2.lacO.CMV.MS2 and imaged at 48 hpt on an LSM900 Airyscan2. Arrowheads indicate multiple actively transcribing VGs in a single episome. **L** U2-OS^LacI-mNeonGreen, MCP-mCherry cells were treated with 1e5 VG/cell AAV2/2.lacO.CMV.MS2 and high-content imaged at 48 hpt. The percentage of LacI foci with at least one overlapping MCP focus in triplicate wells and in biological duplicate are plotted as individual points; each point represents a minimum of 902 imaged nuclei. Error bars = SD. Statistical significance was determined by Kruskal-Wallis Test with Dunn's multiple comparisons test in (**B**, **D**, **G**, & **I**), unpaired, two-tailed *T* tests in (**L**) with *p* = 0.0012, and by two-sided Fisher's exact test in (**C** & **H**), with \**p* ≤ 0.05; \*\**p* ≤ 0.01; \*\*\**p* ≤ 0.001; \*\*\*\**p* ≤ 0.0001. Numbers above graphs indicate fold change over control; black lines = mean values. Source data for plots are provided as a Source Data file. MS2 = Bacteriophage MS2. MCP = MS2 coat protein. mNG mNeonGreen.

inhibition. While Nu7441 has effects on cell cycle (Supplementary Fig. 4A), these effects are too small to be major drivers of the observed transduction effects. This suggests NHEJ positively regulates ITR-ITR joining.

### Blocking HDR increases NHEJ-driven concatenation

Considering the opposing effects on split transgene reconstitution we observed when inhibiting these two main pathways of DSB repair, we hypothesized that after uncoating, HDR factors are preferentially recruited to VGs but are less efficient at concatenation than NHEJ. This suggests that blocking HDR allows NHEJ-mediated concatenation to predominate and thus there is an increase in ITR-ITR joining. To test this hypothesis, U2-OS^LacI-mNG cells were pretreated with B02 or DMSO then transduced with rAAV2/2.lacO.64 array vectors, fixed at 48 hpt and immunostained for Rad51 (HDR) and 53BP1 (NHEJ). High-content imaging revealed Rad51 colocalization with nearly 100% of VG foci imaged in DMSO treated cells, and a dramatic loss of this colocalization in B02 treated cells (Fig. 5G). Concomitantly, B02 treatment increased 53BP1 colocalization from ~10% to >60% of VGs (Fig. 5G).

## Discussion

Effectively overcoming the packaging size constraint would resolve a major limitation for gene delivery with rAAV. The inefficiency of trans-splicing dual vectors has been partially solved by split inteins to circumvent the protein coding limitation, but these still employ strong ubiquitous promoters that offer little control over expression[43]. Elements such as tissue-specific promoters, inducible promoters, enhancer arrays, UTRs/miRNA binding cassettes and other strategies for fine-tuning expression levels and spatiotemporal control must be in *cis* with the ORF they are controlling. Leveraging these approaches to control gene expression therefore necessitates concatenation of multiple VGs. In addition to payload expansion, concatenation is presumed to be an important mechanism for persistence of rAAV delivered DNA and transgene expression[44].

Significantly advancing rAAVs as versatile gene delivery vectors will require a better understanding of the mechanistic underpinnings of rAAV/host interactions. By investigating mechanisms of concatenation and expression from the point of view of the host cell, here we identify druggable targets for increasing dual vector transduction. Taken together, our results propose a model for host-vector interactions and pathway decisions that govern concatenation and expression of dual vectors (Fig. 6). Shortly after uncoating, dsDNA VGs associate with histones and are flagged by the host cell (Fig. 6A). HDR factors such as Rad51 and BRCA1 are preferentially recruited to VGs, which blocks NHEJ factor recruitment, and HDR is the slower concatenation mechanism (Fig. 6B). Blocking HDR allows the more efficient NHEJ to concatenate VGs, resulting in an increase in split

transgene reconstitution (Fig. 6C). Under transient Rad51 inhibition, we achieve dual vector transduction levels comparable to 100-fold higher vector doses without drug. Lowering vector doses is a key step toward ensuring safety of gene therapies. Although prolonged use of DDR inhibitors is dangerous, our results suggest that a single, transient dose of drug preceding vector administration is sufficient to increase transduction, which may be safer to the patient than the toxic responses seen with very high vector doses. Our results also suggest that diseases of deregulated HDR, such as BRCA deficient cancers, may potentially confer an advantage for rAAV clinical applications in some settings.

It is widely believed that the AP fragment in HB dual vectors improves efficiency by promoting HR/HDR between VGs[26,27,45], however this has not been formally tested in this context. Here, we present evidence that HR/HDR inhibits split transgene reconstitution, regardless of AP inclusion. This suggests that AP does not primarily function through HR/HDR. We observed similar effects on TS or HB dual vector transduction when we inhibit HDR or NHEJ (Fig. 5), suggesting that these DSB repair pathways function similarly in ITR-ITR joining or AP overlapping. Upstream of both HDR and NHEJ, the MRN complex recognizes DSB free ends, and has been implicated as inhibitory to wt AAV infection/replication and to rAAV transduction, including rAAV VG foci formation[15–17,35]. The MRN complex may function in multiple ways in response to rAAV VGs, as it does in the response to cellular DSBs.

The vector tools introduced in this study enable us to investigate mechanisms of transduction at individual VG resolution and therefore open new avenues for future studies. Additional research is needed to clarify exactly how Rad51 and HDR impede concatenation of VGs. The canonical role for Rad51 in HR/HDR is to coat ssDNA after resection at the DSB end and promote homology search and strand invasion[46]. One possibility is that the majority of ITRs are in the "T" conformation and are technically dsDNA, and not coated by Rad51. Such ITRs cannot perform efficient homology searches for neighboring ITRs with which they can concatenate. Perhaps this dead-end Rad51 binding impedes NHEJ factor recruitment and by extension concatenation through non-HDR pathways. It is also unclear specifically how Rad51 inhibition increases the proportion of transcriptionally active VGs. Highly transcribed genes are known sites of genomic instability, and how the cell balances expression and repair is an actively researched topic[47,48]. One possibility is that Rad51 is involved in the cellular response to R-loops formed at the strong CMV promoter in our LacO.MS2 array reporter VGs. This rAAV vector system therefore presents a tool for studying fundamental mechanisms of gene regulation and DNA repair.

The LacI-mNeonGreen expressing cell lines have excellent signal to background ratio at VG foci due to low level expression of LacI-mNG. While this enables detection of very small foci early in

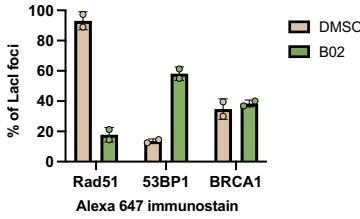

**Fig. 5 | Rad51 inhibition increases dual vector transduction.** The effects of (**A**–**C**) Rad51 or (**D**–**F**) DNA-PK inhibition on split transgene reconstitution were tested in U2-OS cells by LacZ activated fluorescence assay (LAFA) 48 hours post-transduction by the dual (or single) vector indicated above each plot. Bars represent the mean of 5 (**A**–**C**) or 4 (**D**–**F**) biological replicates; individual values are plotted as dots, bars represent SD. A 2-way ANOVA was used, with *$p \leq 0.05$; **$p \leq 0.01$; ***$p \leq 0.001$; ****$p \leq 0.0001$. **G** U2-OSlacI-mNeonGreen cells were pre-treated with DMSO or B02, transduced with 1e5 VG/cell AAV2/2.lacO.64, and immunostained for Rad51, 53BP1, or BRCA1 as listed at top of panel using an Alexa-647 secondary antibody at 48 hpt. **H** LacI and 647 foci colocalization (center of LacI focus overlapping with 647 spot) was quantified by high content imaging in biological duplicate and reported as a percentage of total foci imaged with a minimum of 931 foci imaged per data point (bar represents average of individual plotted points). Source data for plots are provided as a Source Data file. DMSO = dimethyl sulfoxide.

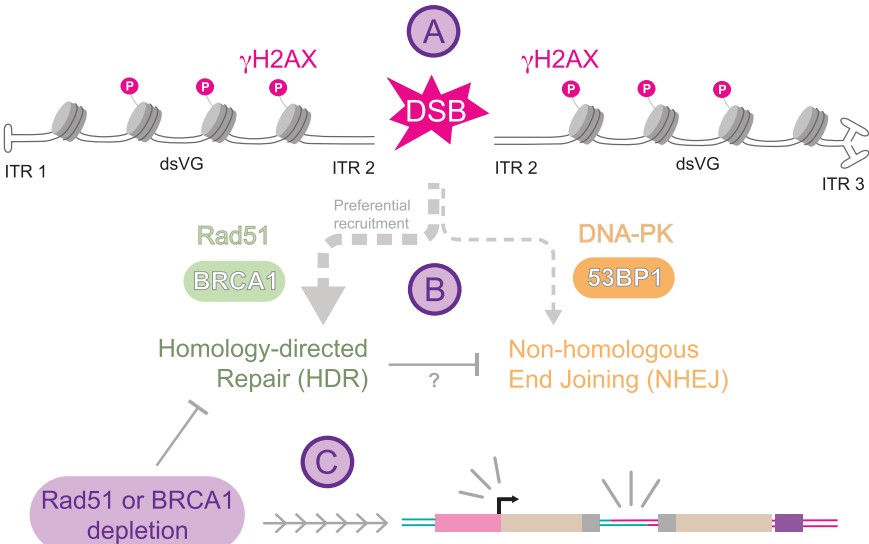

**Fig. 6 | Model for VG processing.** After uncoating, vector genomes (VG) undergo double-strand conversion (dsVG) via two possible pathways that result in at least three inverted terminal repeat (ITR) conformations[41]. ITR-primed second-strand-synthesis results in a T-shaped hairpin at the priming end like ITR 1 and a blunt opposite end like ITR 2, whereas VG-VG anealing produces two blunt (ITR 2) ends, that can also form a double-T structure (ITR 3) if ITRs anneal intramolecularly rather than pairing to the complementary strand as in ITR 2. **A** rAAV vector genomes quickly associate with histones and are marked as double-strand breaks (DSB) via H2AX phosphorylation by the host cell. **B** Homology-directed repair (HDR) machinery is preferentially recruited, which may block successful recruitment of non-homologous end joining (NHEJ) factors. **C** Inhibition or loss of HDR machinery promotes increased expression and concatenation, that synergize to dramatically increase dual vector transduction.

transduction and robust image analysis at all timepoints, it unfortunately limits our ability to quantify terminal sizes of episomes reliably by imaging. The pool of free LacI-mNG in the nucleus becomes depleted as transduction progresses, and it is unclear which, if any, VG arrays are saturated; only saturated VGs can be used to quantify size. Future studies may incorporate clonal lines with mid-range expression levels to enable size quantification under DDR or other pathway perturbations.

Chromatinization and epigenetic regulation are just beginning to be appreciated by the field as major determinants of gene therapy outcomes[49,50]. Epigenetic regulation is a well-known orchestrator of mammalian gene expression or silencing, as well as genome maintenance and repair, but has been only minimally examined in the context of AAV viral and vector genomes contexts. Active[51,52] and repressive[52–54] histone marks have been observed on VGs at late timepoints in transduced cultured cells and in mouse and NHP models. In NHPs, VGs extracted years post injection exist episomally as monomers and high molecular weight concatemers with nucleosomal structure[55]. It is unclear which VGs are expressing versus silenced, but moreover, how and when epigenetic control is exerted on VGs remains understudied. Here, we provide evidence for VG chromatinization very early in transduction in cell culture, and suggest a previously underappreciated role for epigenetic regulation in coordinating concatenation through DSB repair pathways. A limitation of our approach is that it cannot distinguish whether VGs are recruited to host genomic sites of active DNA damage repair and then hijack it, or whether VGs independently recruit these factors. The former has been suggested as the mechanism for wt AAV viral infections[32], but the viral Rep proteins implicated in these studies are absent in the recombinant AAV setting. In wt AAV infections, viral DNA is observed to localize to sites of host damage, where it may capitalize on active repair machinery, however, this behavior is not seen in rAAV VGs[32]. Another limitation is that we cannot determine whether VGs enter and are immediately active or are repressed as the default state, which would require derepression and priming for activation. This is an interesting topic of future study.

rAAV VGs remain predominantly episomal, but integrations can occur at low levels and are of concern for clinical applications. While the present study does not examine integration of VGs, the effects of HDR/NHEJ manipulation on integration are an exciting avenue for future investigation, particularly in the context of rAAV-delivered HDR donor templates for CRISPR-mediated editing or knock-in strategies. The tools introduced in this study will be useful for studying vector-host interactions driving many aspects of rAAV biology as well as fundamental cellular and molecular biology.

## Methods
### Plasmids
*Trans* rep-cap pAAV2/2 (#104963), *cis* pAAV.CMV.Luc.IRES.EGFP.SV40 (#105533) and Helper plasmids pAdDeltaF6 (#112867) were obtained from Addgene. ITR2.LacO.64 array *cis* plasmids were cloned from a pENTR 16.4 LacOR Tjian Lab vector (array of 16 lacO sites concatenated 4 times) into the ITR2-containing backbone from pAAV.CMV.EGFP.Luc. To create ITR2.LacO.64.CMV.mScarlet, The CMV promoter and mScarlet cassette were PCR amplified adding a peroxisome targeting sequence (SKL) to the C-terminus; the amplicon was ligated into RE digested ITR2.lacO.64. ITR2.LacO.64.CMV.MS2 was generated from RE digest to remove mScarlet from ITR2.LacO.64.CMV.mScarlet and RE digest of pMBSV5 (Singer Lab), to yield a fragment containing 24 repeats of the MS2 stem-loop sequence, which was ligated into the ITR2.LacO.64.CMV backbone. All plasmid stocks were propagated in StbL2 competent cells, with some intermediate cloning steps carried out in DH5alpha.

### Vector production and purification
Vectors were produced by triple-plasmid transfection in adherent HEK293T cells. For large-scale production and purification, at least ten 15 cm dishes of HEK293T were transfected with the appropriate plasmids and cells + media were collected 72 h later and subjected to 3 freeze/thaw cycles. Lysate was clarified by centrifugation at 15,000 x g, Benzonase treated for 30 min at 37 C, then lysate was 22 µ filtered and purified by AVB Sepharose Hi-Trap prepacked columns following the manufacturer's instructions on an AKTA pure FPLC system. All titers were assayed by TaqMan qPCR using primers and probes within the transgene cassette, or in the case of LacO.64 which consists only of the repeat array, the ITRs.

## Vector genome quantification in transduced cells

Transduced cells were trypsinized and pelleted, then nuclei were isolated by resuspension in 15 mM Tris HCl, pH 7.5, 300 mM sucrose, 60 mM KCl, 15 mM NaCl, 3 mM MgCl2, 0.5 mM DTT, 1x cOmplete mini protease inhibitor cocktail, and 0.5% NP-40. Cells were allowed to swell on ice, then dounced 10 times with a tight pestle. Nuclei were pelleted at 500 g for 5 min, washed once with the same buffer, then resuspended in PBS + 0.3% SDS and 1 mM EDTA. Lysed Nuclei were incubated with 200 ug RNase A at 37 C for 1 h, then with 40 ug proteinase K at 55 C overnight. Total DNA was phenol-chloroform extracted, and ethanol precipitated, then resuspended in an appropriate volume of 10 mM Tris pH 8. VGs were quantified in equal quantities of total DNA per sample by TaqMan qPCR with primers and probes against the CMV promoter within the transgene cassette and a standard curve of linearized plasmid containing this VG.

## Chromatin Immunoprecipitation

ChIP was performed as described with few modifications[56]. U2-OS cells were transduced with 1e5 VG/cell AAV2/2.Lac0.64.CMV.mScarlet or Empty vector. 10-12 or 40-48 hours after transduction cells were cross-linked 6′ at room temperature with 1% formaldehyde-containing FBS-free medium; cross-linking was stopped by adding PBS-glycine (0.125 M final). Cells were washed twice with ice-cold PBS, scraped, centrifuged for 10′ at 1000 x g and pellets were flash-frozen. Cell pellets were thawed and resuspended in cell lysis buffer (5 mM PIPES, pH 8.0, 85 mM KCl, and 0.5% NP-40, 1 ml/15-cm plate) w/ protease inhibitors and incubated for 10′ on ice. Lysates were centrifuged for 10′ at 500 x g and nuclear pellets resuspended in 6 volumes of sonication buffer (50 mM Tris-HCl, pH 8.1, 10 mM EDTA, 0.1% SDS) w/ protease inhibitors, incubated on ice for 10′, and sonicated to obtain DNA fragments around 500 bp in length (Covaris S220 sonicator, 20% Duty factor, 200 cycles/burst, 150 peak incident power, 10-12 cycles 30″ on and 30″ off). Sonicated lysates were cleared by centrifugation at 10,000 x g and chromatin (75-250 µg per antibody) was diluted in RIPA buffer (10 mM Tris-HCl, pH 8.0, 1 mM EDTA, 0.5 mM EGTA, 1% Triton X-100, 0.1% SDS, 0.1% Na-deoxycholate, 140 mM NaCl) w/ protease inhibitors to a final concentration of 0.8 µg/µl, precleared with Protein G dynabeads (Invitrogen) for 2 hours at 4 °C and immunoprecipitated overnight with 0.75-2.5 µg of specific antibodies. About 4% of the precleared chromatin was saved as input. Immunoprecipitated DNA was purified with the Qiagen QIAquick PCR Purification Kit, eluted in water and analyzed by qPCR together with 2.5% of the input chromatin (SYBR Select Master Mix for CFX, ThermoFisher Scientific). ChIP-qPCR primer sequences for rAAV or human gDNA controls were as follows:

hGenome_Chr17_for: GCTTTTCGCGAGGACAATTC
hGenome_Chr17_rev: GTGCTTGAACCACCACTTCA
AM_qPCR_CMV_F: CATCTACGTATTAGTCATCGCTATTACCA
AM_qPCR_CMV_R: GAAATCCCCGTGAGTCAAACC

Enrichment of specific antibodies was calculated as ΔCt over input signal.

## Cell lines and KO experiments

HEK293T cells used for vector production were obtained from the UC Berkeley BDS Cell Culture Facility (original source ATCC CRL-3216). To generate U2-OS<sup>LacI-mNeonGreen</sup>, U2-OS cells (UC Berkeley BDS Cell Culture Facility cat# 608) cultured in DMEM (1 g/L glucose, 10% FBS) were transfected with a modified PiggyBac Transposase plasmid (System Biosciences) and a PGK.mNeonGreen-LacI.NLS.IRES.Puro integration plasmid using Fugene6 transfection reagent. After 48 h, cells were subjected to Puromycin selection (1µg/mL) for 96 h. Survivors were expanded for an additional 96 h at maintenance (0.2µg/mL) Puromycin concentration. Individual cells were then plated in 96 well plates using a FACS Aria. After 18 days, individual colonies were expanded to larger wells and to replicate CellCarrier Ultra Plates

(Perkin Elmer) for initial screening on an OperaPhenix high-content imager. Live colonies were imaged with a 488 filter set, and those expressing mNeonGreen within an ideal threshold (nuclear signal between 200-2000) were passaged to replicate Falcon and CellCarrier Ultra plates for additional screening (48 clones total). Chosen colonies were then transduced with 1e9 vg/well rAAV2/2.lacO.64 crude vector preparation and live imaged at 48 hpt. The three colonies with best signal to background ratio at LacI-mNG foci were expanded and used for further experiments. To generate U2-OS<sup>LacI-mNeonGreen.MCP-mCherry</sup> cells, one clonal line was transfected with PiggyBac Transposase and a UBC.tandemMCP2-mCherry plasmid with a neomycin resistance marker integration plasmid. Selection under 1 mg/mL G418 was carried out as above, and clonal expansion and screening performed as above, imaged with mCherry filter sets after transduction with rAAV2/2.lacO.64.CMV.MS2.24. To generate Cas9-expressing lines, U2-OS<sup>LacI-mNeonGreen</sup> or U2-OS<sup>LacI-mNeonGreen.MCP-mCherry</sup> cells were infected with a Cs9/Blasticidin Lentivirus, then cells were put under selection (10ug/mL Blasticidin) for one week, and plated as individual cells in 96well plates. Expanded clones were validated for Cas9 presence and activity, and highly active clonal lines chosen and expanded. For BRCA1 KO experiments, U2-OS<sup>LacI-mNeonGreen::Cas9</sup> cells were transfected in an Amaxa Nucleofector to deliver BRCA1 or Scrambled sgRNA expressing plasmids also carrying a Zeocin resistance cassette. Cells were then selected for 72 h under 1 mg/mL Zeocin, then transduced with rAAV, fixed at 48 hpt, and stained and imaged as described below. The guide sequences are as follows (BRCA1 guide plasmids were pooled at an equimolar ratio):

Scramble: GCACTACCAGAGCTAACTCA
BRCA1:
1. AGAAACCTACAACTCATGGA
2. AGGAAACATGTAATGATAGG
3. GTTTCTATCATCCAAAGTAT

## LacZ-activated fluorescence assay (LAFA)

U2OS or BeWo cells were seeded in DMEM (1 g/L glucose and 10% FBS) at a density of 2,500 cells/well in a 96-well plate. The following day, cells were pretreated with the pharmacological inhibitors B02 (5 M or 10 M) or Nu7441 (1.75 M or 2.75 M) in DMEM (high glucose and 10% FBS) for 4-16 h. Spent media was replaced with DMEM containing the appropriate dose of unsplit or split LacZ vector, and B02 (5 M or 10 M), Nu7441 (1.75 M or 2.75 M), or DMSO, and 5% FBS. Cells were transduced for a total of 48 hours without further media changes. Following transduction, cells were washed with 1x PBS and lysed in plate using 50uL cold lysis buffer (2 mM DTT, 25 mM Tris-phosphate (pH 7.8), 2 mM 1,2-diaminocyclohexane-N,N,N´,N´-tetraacetic acid, 1.25 mg/ml lysozyme, 2.5 mg/ml BSA, 10% glycerol, 1% Triton® X-100). To facilitate lysis, cells underwent two freeze/thaw cycles at −80 C and 37 C, respectively. 4-Methylumbelliferyl-beta-D-galactopyranoside (MUG, 10 mM in DMSO) was diluted 1:10 in LAFA buffer (100 mM Sodium Phosphate Buffer (pH 7.0), 1 mM MgCl2, 10 mM 2-mercaptoethanol, 0.1% Triton® X-100) to a final concentration of 1 mM. In a 96-well plate, 20uL of the cell lysate was added to 100uL of the MUG-LAFA buffer mixture, protected from light, and gently rotated at room temperature for 90 minutes. Fluorescence signal was read at 365 nm/460 nm (Ex/Em) using a Tecan Infinite® M1000. To control for collecting LAFA signal from equivalent numbers of cells, ATP was quantified using a CellTiter-Glo® kit according to the manufacturer's protocol, and this was used to normalize per well fluorescence.

## siRNA knockdown experiments

siGENOME SMARTPool siRNAs were purchased from Horizon Discovery for Human RAD51 (cat# M-003530-04-0010), Human BRCA1 (cat# M-003461-02-0010), and Non-targeting siRNA control pool #2

(cat# D-001206-14-20). For transfection into cells, Lipofectamine® RNAiMAX was used according to the manufacturer's instructions. At appropriate time post-transfection, cells were either collected for western blot or flow cytometry, or transduced with rAAV in fresh media for imaging experiments.

## Western blotting

Cells in 6-well plates were drug or siRNA treated as indicated. Wells were washed with PBS, 250 μL RIPA buffer added, plate was freeze/thawed once at −80, wells were scraped on ice, and lysate collected and clarified at 15000 g for 10 min at 4c. Total protein was quantified by Pierce™ 660 nm Protein Assay kit according to mfr instructions. 12 ug total protein was diluted to 1x in SDS PAGE buffer and loaded in each well of a NuPAGE 4-12% Bis-Tris gel and electrophoresed proteins transferred to a nitrocellulose membrane. The membrane was blocked in PBS + 0.1% Tween 20 (PBST) + 5% nonfat dry milk for 1 h at RT, then primary antibody was diluted in PBST + 0.5% milk and incubated overnight at 4 C. Membrane was washed twice in PBST at RT, secondary antibody diluted in PBST + 0.5% milk and rocked for 1 h at RT, then washed twice in PBST and imaged on a ChemiDoc.

## Screen and pathway-enrichment analysis

A lentiviral Brunello/Cas9 library was purchased from the Broad Institute Genetic Perturbation Platform (GPP) and used to infect 400 M U2-OS cells at an MOI of 0.5. Puromycin selection began 48 h later and continued for 96 h. The surviving cell library was passaged once into 15 cm plates and 40 M cells were transduced with 5e3 vg/cell rAAV2/2.HB L and R vectors in biological duplicate. At 48 hpt, cells were collected and stained with MarkerGene™ (a commercially available MUG analog kit) according to the manufacturer's protocol, sorted on a FACS Aria (BD), and gDNA extracted from each population with the Qiagen DNeasy Blood and Tissue kit according to manufacturer's protocol. NGS was performed at the Broad GPP core facility.

Log-normalized sgRNA barcode counts from FACS-sorted MUG-negative cells were subtracted from log-normalized sgRNA barcode counts from FACS-sorted MUG-positive cells. This difference was used to model a hypergeometric distribution using the Broad GPP Web Portal (https://portals.broadinstitute.org/gpp/public/). From this analysis, we recovered enrichment scores and *p*-values associated with individual sgRNA barcodes, which we mapped to gene perturbations. Genes with fewer than 2 unambiguous sgRNA perturbations were culled from subsequent analyzes. Plots based on these statistics were made using the pandas and matplotlib libraries in Python.

The most significant 5% of positive regulators (i.e., genes that, when perturbed, have increased dual vector transduction) were submitted as a gene list for Reactome pathway analysis[57]. Pathways significantly overrepresented within this gene list, and their respective *p*-values, were used to make matplotlib plots and inform hit validation.

Genes in the top 5% of negative regulators grouped by their Reactome pathway(s) (multiple pathways can contain the same gene) are tabulated with their fold-enrichment scores, and their *p*-values (Supplementary Data 1). Code used for operations on tables, to make supplementary tables, and to plot analyzes can be found at https://doi.org/10.5061/dryad.7h44j1053 and are included as a Jupyter notebook (Supplementary Data 2).

## High-content imaging and analysis

Cells were plated in CellCarrier Ultra 96 well plates (Perkin Elmer) and transduced/treated with the appropriate drugs and vectors. Cells were fixed and stained with Hoechst to facilitate nuclear segmentation, and immunostained where indicated with the below primary antibodies overnight at 4 C for detection by an Alexa647-conjugated secondary antibody. ROIs comprising up to 50% of the well surface were imaged with 11 z-planes at 0.5 micron steps using the appropriate filter sets on an Opera Phenix automated imager (Perkin Elmer). MIPs were processed and quantified using Harmony (v5.1) image analysis software using an appropriate analysis pipeline (Supplementary Data 2). Well and object results were imported into Graphpad Prism (v10.1.0) for data visualization.

## Cell cycle analyzes

For analyses by flow cytometry, cells were seeded in 12-well plates and treated with drug or transfected with siRNA 24 h later. At 40 or 36 h post treatment, cells were trypsinized and pelleted, resuspended in Phenol Red-free DMEM + 0.01% digitonin + 2.5 μg/mL DAPI, and 5000 cells per condition were analyzed by flow cytometry on a LSR Fortessa and FACSDiva software (v 9.0) from BD Biosciences. Gating for G1, S, and G2/M phases was performed on DMSO treated samples. Data were analyzed using FlowJo software (v 10.9.0).

For imaging studies of S-phase, cells were transduced with 1e5 vg/cell of AAV2/2.LacO.64.CMV.mScarlet for 48 h, then media changed to DMEM + 10 μM EdU + 10% FBS for 15 minutes. Cells were then fixed and EdU detected using the Click-iT™ EdU Cell Proliferation kit (Alexa 647) according to the mfrs instructions. Cells were then immunostained and imaged as described in High-Content Imaging and Analysis.

## Antibodies and dilutions

**Immunofluorescence.** Rb anti-53BP1: Abcam ab175188, lot# GR135905-4, 1:10,000. Rb Phospho-Histone H2A.X: Cell Signaling Technology 25775, lot# 14, 1:2500. Ms anti-BRCA1: Santa Cruz sc-6954, lot# J1821, 1:2000. Rb anti-Rad51: Axxora CAC-BAM-70-001-EX, lot# 04, 1:4000, and ab133534, lot# GR3270300-18, 1:2500.

**ChIP-qPCR.** Normal rabbit IgGs: Abcam #ab46540, 0.75 μg per ChIP. Rabbit anti-Phospho-Histone H2A.X (Ser139): Cell Signaling Technology #2577, lot# 14, 2.5 μg per ChIP. Rabbit anti-Histone H2B: Invitrogen # MA5-24697, 1.5 μg per ChIP.

**Western blotting.** Rb anti-Rad51: ab133534, lot# GR3270300-18, 1:2000. Ms anti-BRCA1: Santa Cruz sc-6954, lot# J1821, 1:1000. Ms anti-TBP: Recombinant Anti-TATA binding protein TBP antibody [mAbcam51841] (ab300656), lot# GR135905-4, 1:2500.

## Live imaging experiments

Cells were seeded in ibidi Culture-Insert 4 Well μ-Dishes (cat# 80466), pretreated with DMSO or B02 for 8 h, then 1e6 vg/cell was added to wells and imaging commenced 4 h later on a Nikon W1 SoRa spinning disk with humidified CO2 incubation chamber and the NIS Elements software (v 9.0). A 10-micron stack of images were acquired every 6 minutes for 48 consecutive hours. Maximum Intensity Projections (MIP) were generated from these stacks, and MIP movies were generated in ImageJ (v 2.14.0). After appropriate cropping, substacks were generated and processing to stabilize nuclei in the field of view was accomplished using the StackReg plugin on the Rigid Body setting.

## Reporting summary

Further information on research design is available in the Nature Portfolio Reporting Summary linked to this article.

# Data availability

Raw sequencing data from the genome-wide screen are available at NCBI Sequencing Read Archive under BioSample accession number PRJNA1195613. Source data are provided with this paper.

# Code availability

Code used for operations on tables, to make supplementary tables, and to plot analyzes can be found at https://doi.org/10.5061/dryad.7h44j1053 and are provided as a Jupyter notebook (Supplementary Data 2).

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

## Acknowledgements

We thank Eva Andres-Mateos, Heikki Turunen, Mohammadsharif Tabebordbar, Thomas Graham, Erin Merkel, Nam Che, John Doench, Eric Zinn, Dirk Hockemeyer, Antonio Maffia, Namrata Kumar, and Tyler Huycke for helpful discussions. Thanks to Mathieu Nonnenmacher for critical reading of the manuscript. We thank Rob Singer and Luk Vandenberghe for plasmids. Thanks to the Genetic Perturbation Platform (Broad Institute) for assistance with processing screening samples. We especially thank Mary West of the High-Throughput Screening Facility (HTSF) at UC Berkeley. This work was performed in part in the HTSF, that provided the OperaPhenix high content imager. Funding was provided by the California Institute for Regenerative Medicine Training Program (EDUC4-12790, A.C.M.), the National Institute of Allergy and Infectious Diseases (R21AI185720, M.D.W.), and the Howard Hughes Medical Institute (34430, R. T.). Additional support for the High-content imaging experiments was provided by the UC Berkeley HTSF (NIH 1-U54CA231641-01, UCB).

## Author contributions

A.C.M. Conceived the study and designed the experiments. A.C.M., B.B., and G.M.D. designed and produced recombinant reagents (viral vectors and DNA). A.C.M., B.B., O.N.W., C.C., and D.U.K. performed experiments and collected data. V.B.F. wrote code to visualize screen data. A.C.M., B.B., O.N.W., V.B.F., C.C., D.U.K., and M.D.W. analyzed and interpreted data. A.C.M. prepared the manuscript; All authors discussed the results and implications and commented on the manuscript at all stages. M.D.W., R.T., and X.D. supervised the work. A.C.M., M.D.W., and R.T. secured funding.

## Competing interests

The authors declare no competing interests.
