## [Transparent Peer Review file · Nature Communications]

Double-Strand Break Repair Pathways Differentially Affect Processing and Transduction by Dual AAV Vectors

Corresponding Author: Dr Anna Maurer

Version 0:

Reviewer comments:

Reviewer #1

(Remarks to the Author)

In this study, Maurer et al. report a promising finding that could greatly improve large gene delivery in human cells. Recent engineering of adeno-associated viral vectors (rAAV) revealed the potential of using concatenation of viral genome to double the delivery size of the gene to be transduced. Using a cleaver fluorescence-based genome-wide CRISPR-Cas9 screen approach and flow cytometry analyses, the authors investigated the cellular component that impacts rAAV concatenation. Surprisingly, genes essential for DNA repair by homologous recombination repair emerged as important repressors of rAAV-mediated gene transduction. Based on the observation that viral genomes colocalize with gH2AX, the authors conclude that concatenation of rAAVs triggers a DNA double-strand break (DSB) response on chromatinized viral genome. Deleting the DNA repair factors BRCA1 and inhibiting the ATPase activity of the recombinase RAD51 efficiently increase viral transduction (concatenation and gene expression from the viral genome), validating the hits identified in the original screen. Consistent with Non-Homologous End Joining (NHEJ) repair promoting viral concatenation, inhibition of the protein kinase DNA-PKcs significantly reduces viral transduction.

This study provides important findings that would be of interest for improving the rAAV-based technology for transgene delivery and the biology of the viruses, that is, viral genome chromatinization and concatenation. However, the current version of the manuscript is missing important controls and information needed for publication in Nature Communications.

Major:

1- It is not clear why the authors conclude that the concatenation triggers DSBs. First, the authors should highlight which structure of the viral DNA could be recognized as DSBs (this is not clear in the models provided). Second, the histone variant H2AX can be phosphorylated by ATM upon the detection of DSB but also by ATR upon replication stress. This possibility could be verified by testing the impact of ATM and ATR inhibitors on viral concatenation/transduction efficiency. Third, the authors should provide quantification of the colocalization between the viral genome, BRCA1 and 53BP1. 53BP1 accumulates at DSB that accumulates in the cellular genome in the S-phase. Can the authors speculate why it would be excluded from the viral genome? This phenomenon is even more intriguing, considering that NHEJ factor DNA-PKcs positively impact concatenation. Are most of the cells with BRCA1 colocalization in the S-phase? This point could be addressed using an antibody against PCNA or EDU incorporation to mark cells in the S-phase. And why not look at the accumulation of DNA repair protein on the viral genome 4 hpt as it is done for gH2AX?

2- Concluding that the viral genome is chromatinized based on the colocalization of gH2AX is a strong statement. Can the authors use their Lacl construct to pulldown viral DNA and validate nucleosome incorporation?

3- The conclusion in figures 3 and 4 assumes that the foci detected using the Lacl construct are concatenated viral genome. The Lacl construct binds to viral genomes that have been converted to dsDNA. Can the authors rule out the possibility that the foci detected by microscopy are monomers of the viral genome? Does Lacl binding impact the efficiency of concatenation?

4- In Fig 3, can the author comment on the nature of the foci observed in the DAPI channel?

5- Important information is missing in the material and method (the method used to quantify the viral genome presented in fig

4I, details on the information presented on the graphs – mean vs median. The constructs used for the CRISPR screen contain an mCherry expression cassette as well as a blasticidine and a GFP expression cassette that are dependent on concatenation. Have these elements been used at any point during the screen? How was the BRCA1 KO validated? Is this a clonal cell line? How was the inhibition of RAD51 confirmed?

Reviewer #2

(Remarks to the Author)

In the study conducted by Maurer et al., it was observed that homologous recombination (HR) exerts an inhibitory effect on dual vector transduction. Through the implementation of a genome-wide gene KO screen, the authors illustrated that the depletion or inhibition of HR factors, such as BRCA1 and Rad51, leads to a significant enhancement in the reconstitution of a large split transgene. This improvement is attributed to an increase in both concatenation and expression from rAAV. While the manuscript employs innovative methods to unravel the biological processes of rAAV, it is important to note that certain pivotal experiments are necessary to rule out alternative hypotheses before definitive conclusions can be drawn.

Specific points:

In Fig. 2B, the authors demonstrate the widespread use of genome-wide CRISPR screening in various studies. While RAD51 and BRCA1 are mentioned as significant hits, it would enhance reader comprehension if the authors provided a comprehensive list of all enriched genes in Fig. 2B. Additionally, presenting this information in a table with fold changes and enrichment scores for each gene would be valuable.

In Fig. 3B, the visualization reveals limited colocalization.

In lines 182-186 and Fig. 4, prior to the conclusions drawn in lines 185-191, it is recommended to include Western blot validation of BRCA1 gene knockout efficiency. Given that BRCA1 is a DNA repair-associated protein and DNA damage is reported to be crucial for rAAV transduction, assessing whether BRCA1-KO affects cell cycle or DNA damage response, potentially influencing rAAV transcription, would be insightful.

In lines 191-192, it would strengthen the manuscript to provide direct evidence, such as qPCR or NGS data, demonstrating the inhibition of concatenation and de novo episome formation.

In Fig. 4E-G, a Western blot is essential to validate the effect of Drug B02. Considering RAD51's significance in this study, it is advisable to include RAD51 knockout or knockdown experiments to further validate RAD51's function.

In Fig. 4J-K, clarification is needed regarding why colocalization was not observed.

In Fig. 5, it is suggested to include Western blot analyses, cell viability assays, and cell cycle assessments to validate the inhibitory effects of B02 and NU7441. Furthermore, Fig. 5G should incorporate images to visually demonstrate the relevant findings.

Reviewer #3

(Remarks to the Author)

Summary

One of the critical drawbacks to the use of recombinant AAV gene therapy vectors is the limitation of the transgene size, precluding the possibility of using therapeutic transgenes that exceed the packaging capacity of the AAV capsid. This drawback can be overcome by using multiple rAAV vectors that are engineered to recombine in the host cell and reconstitute a large therapeutic transgene. However, these rearrangements have place at low efficiency. Little is known about how rAAV genomes recombine to form extrachromosomal concatemers and how they express. To address this deficiency in knowledge, in this manuscript, Maurer et al. have investigated the mechanisms of homologous recombination of rAAV vectors, dissecting how both concatemer formation and expression from rAAV vector genomes are related, to improve the efficiency of dual vector systems. Using a split dual vector system, the authors perform a CRISPR screen to identify the host factors mediating concatenation of rAAV genomes in the host nuclei. These assays identified the homologous recombination pathway of DNA repair as the mechanism facilitating rAAV rearrangements. Using a previously developed rAAV vector system containing LacO binding sites to label rAAV genomes, they determine how localization and distribution are impacted upon inhibition of the HR pathway. These discoveries are exciting for the field of gene therapy. While these findings are novel and of interest, the authors have made some conceptual and proposed mechanistic leaps that are not corroborated by experimental evidence. Claims regarding how these essential DNA repair proteins can be targeted by pharmacological intervention to improve rAAV transduction disregard detrimental effects to patients and are not necessary. These discrepancies should be addressed with corrections to their model or appropriate experimental evidence, which would further strengthen their conclusions. The specific suggestions for improvement are detailed below:

Major Comments

Line 113 – 115: The authors should provide more clarity on the difference between the dual vector systems used in the

experiment. How is the reconstituted transgene efficiency affected by the additional AP segments in the Hybrid vectors? Do the trans-splicing (TS) vectors need validation from the Hybrid vectors? Were the cells infected with only one set of dual vectors at a time? These experiments and appropriate controls need to be described more fully.

Figure 3B: An empty vector only control is needed for this experiment. Why is the gH2AX timecourse assay not between 4-48 hours as shown with previous experiments to show initial gH2AX recruitment and disassociation to initiate transcription? Because transcription cannot occur on packaged in gH2AX, the authors should provide some sort of explanation for these observations. RNA pol II should be utilized as a control. Authors also need a positive control for DNA damage (such as using HU or UV radiation).

Line 165 – 166: The authors should provide appropriate references and experimental evidence to support their claim that the vector genomes are chromatinized immediately after uncoating and are subsequently flagged as DSBs with appropriate timecourses. gH2AX is a marker for both host and viral DSB. Could this mean that rAAV is inducing double-stranded breaks on the host genome? Although the LacI-mNeon-Green is a great system to label rAAV vectors it does not indicate concatemerization to the same certainty at the Split donor and acceptor vectors. This drawback makes the LacI-mNeon-Green vector an inadequate system for measuring recombination.

Line 180: The authors should provide experimental evidence that increase in the size of the mNeonGreen foci are associated with concatenation.

Figure 4: The overall findings of Figure 4 require controls to show that the findings are not caused by derepression of the rAAV genome. The difference between Fig. 4B and 4C are unclear. The authors should provide error bars and statistical significance in 4C. These experiments should be accompanied by a BRCA1 western blot to confirm the knockout. Independent gRNAs are needed for the CRISPR knockout studies. The graph in Fig. 4I seems meaningless because it is not compared with any treatment conditions.

Figure 5G: The authors should include representative images of VG foci with error bars and statistics. This experiment should also be done with DNA-PK inhibitor NU7441 to measure the impact of vector transduction/recombination upon NHEJ signaling.

Figure 6. This diagram shows gH2AX binding to the double-stranded rAAV genome. It has not previously been convincingly established by chromatin immunoprecipitation assays that gH2AX recognizes the rAAV vector as a double-strand break.

Minor Comments

Figure 1B-D: It is unclear if RNA processing proceeds or follows DNA recombination. How does this assay show more than two separate rAAV vectors can recombine? Control for rearrangement of the splice donors as transcripts (before actual recombination of the rAAV vectors).

Figure 1E: This graph is uninformative and colors are confusing. What is the rearrangement frequency shown in this graph? Would it be possible to depict this in the form of a 3D plot for more clarity?

Line 140: The authors refer to a pool performed to a generated list of candidate inhibitors for dual vector transduction. Please include hits from that screen. Authors need to clarify which genes they knocked out in this screen.

Line 142: Authors need to clarify the pathway analysis of significantly enriched genes taken from the negative or positive hits from the screen. If they are taken from the MUG negative it will indicate that recombination is inhibited by the CRISPR knockouts and the opposite for the MUG positive.

Figure 2B: Representative dots of RAD51 and BRCA1 on the waterfall plot should be more prominent in color. What is the cutoff of the waterfall plot? What is the meaning of positive and negative population and how does it correlate to the negative and positive number demarcations?

Figure 2C: This figure is uninformative and gene regulation categories need to make more sense. How is the "Cell cycle" category different from the "Cell cycle, Mitotic"? What is the overlap between these categories what genes would fit in one and not the other? What are some example genes that have been identified in the screen besides RAD51 and BRCA1 and how do they fit in each category? The authors should consider taking out this figure or focusing it only on the HR and NHEJ genes.

Line 183 Authors need a reference to the BRCA1 scrambled guides used in the U2-OS cells.

Line 191 – 192: The observation of Neon Green foci increase has been established to represent concatenation or episome formation (line 162) which seems to contradict the statement that "BRCA1 inhibits concatenation." However, imaging showed a significant increase in the average number of VG (Neon Green) foci per cell. How does this data represent an inhibition of concatenation? A similar contradiction can be found in line 203 with RAD51 inhibition.

Line 237 – 238: Authors should not say this without disclosing how chemical inhibition of RAD51 or BRCA1 can affect the host cell and cause DNA damage and cancer.

Line 241 – 251: In this section, although the author's purpose seems to be to connect the dual vector as a model for concatenation to HR and HDR seems unnecessary and should be moved to the discussion.

Figure 5A-5F: Is the dual vector system driving Homologous recombination or is driven by AAV recombination? In lines 251-253, the authors sound like they are looking at the mechanisms of the split donor's system and not rAAV biology. Authors should include fluorescence threshold in all graphs. They should also explain (or speculate) why there is a decrease in fluorescence when more Vector dose is added.

Line 261: The authors need to provide references to support their assertion that concatenation can be considered analogous to repairing chromosomal DSB.

Version 1:

Reviewer comments:

Reviewer #1

(Remarks to the Author)

All the issues raised in the original manuscript have been addressed. It is noted that the authors put important effort into providing additional data to support their hypothesis, strengthening the revised manuscript.

The outcome of testing ATRi is quite intriguing and raises interesting questions about the potential role of this kinase in inhibiting transgene expression.

Reviewer #2

(Remarks to the Author)

The studies led by Maurer AC, et al. carried out a genome-wide gRNA knockout screen in U2-OS cells and identified the homologous recombination (HR) pathway plays a negative role to the genome recombination-based dual AAV vector transduction. They showed inhibition of HR-critical factor BRCA1 and Rad 51 significantly increased the transduction of AAV-dual vectors. In the revised version, while the authors address several technical questions, the study still has limitations regarding the understanding of AAV vector transduction biology and its applications.

Major points.

1. All the experiments were performed in U2OS cell lines. It is a human osteosarcoma cell line. The cells are highly proliferating and widely used for study of DNA damage and repair pathways. While both HR and NHEJ are available in U2OS cells, have the authors examined the function of the HR factor BRCA1 and Rad 51 in the transduction of the dual AAV vectors of other proliferating cell line cells?

2. While HR generally prefers high fidelity repair during the S and G2 phases of the cell cycle of the proliferating cells, and NHEJ being used in other phases of the proliferating cells, and more importantly in non-proliferating cells/differentiated cells, when the HR pathway is limited or not available. Initially, the dual AAV vector-based trans-splicing was found and applied in muscle cells, where the recombination-based dual AAV vector transduction was high. Muscle cells are differentiated cells. In differentiated cells, where most of the AAV-based gene therapy applied, the HR machinery is not available. How do the authors address this reality?

3. Please check if the NGS data of the lentiviral Brunello/Cas9 library screen have been deposited to the NCBI (Sequencing Read Archive (SRA)/BioProject numbers were not found in the manuscript).

Minor: Line 127 lacks references

Reviewer #3

(Remarks to the Author)

In this study, Maurer et. al. has exploited the natural concatenation property of rAAV vectors to generate large rAAV transgenes that are larger than the packaging capacity of the capsid. Leveraging this dual vector system, they have performed a CRISPR screen, identifying the homologous recombination (HR) pathway as the principal signaling pathway responsible for regulating rAAV concatenation and subsequent gene expression. Importantly, inhibition of HR proteins leads to increase in concatemer formation and expression of the split transgene, suggesting a transient dosage that might be applicable in-vivo for expression of the split transgenes in patients. Taken together, these studies identify a critical DNA repair pathway that is required for regulating the formation of rAAV concatemers in the host. These studies are sure to spur exciting discoveries in engineering gene therapy vectors that exceed the capacity of one capsid and also provide insights into how viral episomes persist in the nuclear environment. In this resubmission, the authors have gone to great lengths to provide detailed responses to all the reviewer's comments with extensive additional experiments, control studies and previously published data from the literature that validate their findings. These findings have significantly strengthened the overall conclusions of the study and are an important contribution to the field of vector biology. The overall conclusions and claims are supported by experimental evidence, the author's methodology is sound and meets the expected standards of the gene therapy and DNA damage fields. The authors have provided enough detail for the methods of the work to be reproduced.

RESPONSE TO REVIEWER COMMENTS

We wish to thank the reviewers for their time providing these valuable comments, and for their patience with us addressing them after an unexpectedly long but required leave of absence for the first and corresponding author. We are pleased that all reviewers support the value of our study. Accompanying these responses to their comments in the revised version of the manuscript, we have highlighted all additions to the main text in yellow, have made substantial additions to the main figures, added five new supplemental figures/tables, and six new movies. We are grateful for how much the revisions have strengthened the manuscript.

Reviewer #1 (Remarks to the Author):

In this study, Maurer et al. report a promising finding that could greatly improve large gene delivery in human cells. Recent engineering of adeno-associated viral vectors (rAAV) revealed the potential of using concatenation of viral genome to double the delivery size of the gene to be transduced. Using a cleaver fluorescence-based genome-wide CRISPR-Cas9 screen approach and flow cytometry analyses, the authors investigated the cellular component that impacts rAAV concatenation. Surprisingly, genes essential for DNA repair by homologous recombination repair emerged as important repressors of rAAV-mediated gene transduction. Based on the observation that viral genomes colocalize with γ H2AX, the authors conclude that concatenation of rAAVs triggers a DNA double-strand break (DSB) response on chromatinized viral genome. Deleting the DNA repair factors BRCA1 and inhibiting the ATPase activity of the recombinase RAD51 efficiently increase viral transduction (concatenation and gene expression from the viral genome), validating the hits identified in the original screen. Consistent with Non-Homologous End Joining (NHEJ) repair promoting viral concatenation, inhibition of the protein kinase DNA-PKcs significantly reduces viral transduction.

This study provides important findings that would be of interest for improving the rAAV-based technology for transgene delivery and the biology of the viruses, that is, viral genome chromatinization and concatenation. However, the current version of the manuscript is missing important controls and information needed for publication in Nature Communications.

Major:

1- It is not clear why the authors conclude that the concatenation triggers DSBs. First, the authors should highlight which structure of the viral DNA could be recognized as DSBs (this is not clear in the models provided).

We apologize that we were not clear enough, and wish to clarify that we do not conclude that concatenation triggers DSBs; our conclusion is that VGs are recognized as DSBs and then become concatenated. We have carefully edited the text to make sure this is clear. Please note that we state the latter starting at line 377. To clarify the schematic in Figure 6 and address the reviewer's concern, we have redrawn the post double-strand conversion vector DNA more precisely at its ends to illustrate all possible ITR conformations. We have also revised the figure legend to explain this point and added a reference where these are described. The proportion of ITRs per conformation in living cells is not known, but considering the differential conformations provides clarity to the model.

Second, the histone variant H2AX can be phosphorylated by ATM upon the detection of DSB but also by ATR upon replication stress. This possibility could be verified by testing the impact of ATM and ATR inhibitors on viral concatenation/transduction efficiency.

We are grateful to the reviewer for highlighting this point and have added additional experimental data to address their concern. We examined the effects of adding inhibitors to ATM and ATR as suggested by the reviewer and these data are included in Supplemental Figure S3 F-J. We have modified the main text as follows (lines 266-281)

ATM and ATR inhibition increase transgene expression

The histone variant H2AX can be phosphorylated by ATM upon DSB detection, or by ATR upon replication stress. To examine this, we pharmacologically inhibited these kinases (Figure S3 F-J). We observe a decrease in γ H2AX/VG colocalization under ATM inhibition, whereas ATR inhibition has no effect (Figure S3F). This suggests that ATM

phosphorylates H2AX associated with VGs. It should be noted that ATM inhibition decreases global γ H2AX staining and ATR inhibition does not (Figure S3G). Consistent with previous reports^{37,38}, compared to DMSO controls we observed an increase in transgene expression under ATM inhibition (Figure 3H). ATR is activated upon wt AAV2/Adenoviral coinfection³¹ and UV-inactivated AAV2 monoinfection³⁹, and ATR inhibition has been shown to decrease wt AAV DNA replication³¹, but ATR is not well-studied in the rAAV transduction setting. We observed a large increase in transgene expression under ATR inhibition (Figure S3H). Conversely, these drugs have little effect on the number of LacI foci per nucleus (Figure S3I), and ATR inhibition, but not ATM inhibition, significantly increases the percentage of cells with any LacI foci (Figure S3J). The large effects of ATR inhibition on transduction may not depend on γ H2AX but may be due to proteomic changes that affect HR in these conditions⁴⁰.

Third, the authors should provide quantification of the colocalization between the viral genome, BRCA1 and 53BP1.

We had already quantified 53BP1 and VG colocalization (previous figure 5G) and have now added BRCA1 quantification per the reviewer's request. Note that we have moved all images from the 48hpt timepoint to figure 5G-H, and figure 3 now contains only 4hpt images in response to the reviewer's comment below.

53BP1 accumulates at DSB that accumulates in the cellular genome in the S-phase. Can the authors speculate why it would be excluded from the viral genome? This phenomenon is even more intriguing, considering that NHEJ factor DNA-PKcs positively impact concatenation.

It was not our intention to suggest that 53BP1 is completely excluded from VGs, only that it was less frequently colocalizing. The additional primary images and quantification that the reviewer requested has indeed helped to clarify this (Figure 3 D-E). A plausible hypothesis for why HR machinery is preferentially recruited is that the VG is mostly single-stranded upon uncoating, and may readily recruit ssDNA binding proteins such as Rad51 as a result.

Are most of the cells with BRCA1 colocalization in the S-phase? This point could be addressed using an antibody against PCNA or EDU incorporation to mark cells in the S-phase.

The reviewer raises an interesting point. To examine this, we pulsed transduced cells with EdU for 15 min before fixing, labeling EdU with 647 Azide, and staining for BRCA1 (Figure S4 C-H). There may be a slight increase in the percentage of VG colocalizing with BRCA1 in S-phase cells (panels F-H), and this makes sense because BRCA1 is upregulated in this phase. More BRCA1 foci are indeed detectable in panels F-H, so statistically there may be more BRCA1 colocalization, but there does not appear to be a very significant difference. Panel G is an example with a very high number of VG foci (16), and 6 are colocalizing with BRCA1 (37.5%), whereas panel C the cell is not in S-phase and has 2 of 6 foci (33.3%) colocalizing. Panel G is an extreme example – we also see cells with few colocalization events in S-phase (panel F) as well as no foci detectable above background by the image analysis software (panel H). We would like to note that there is no obvious difference in S-phase cells versus other phases in the number of foci per cell or frequency of cells with any foci. We cannot perform high content quantification analysis on these experiments, since we used all four laser lines with different wavelengths to detect the VG (488) and its mScarlet reporter (555) plus EdU (647) and anti-BRCA1 secondary AB (405), and we could not reliably segment nuclei in the analysis software without a Hoechst stain. Since cell cycle genes were also enriched in the screen, we will certainly examine S-phase more thoroughly in our future studies, but this is outside the scope of the current manuscript. We have summarized this in the main text (lines 260-264).

And why not look at the accumulation of DNA repair protein on the viral genome 4 hpt as it is done for γ H2AX?

We thank the reviewer for this excellent suggestion which has strengthened one of our main conclusions of this manuscript. We have added quantification of LacI-mNG, BRCA1, and 53BP1 at 4hpt (Figure 3E) with primary images (Figure 3D) showing examples of co-staining patterns. Colocalization of both repair factors were observed with VG foci, with BRCA1 more often than 53BP1. We quantified overlap of these foci in 957 nuclei (Figure 3E); BRCA1 colocalizes with 72.4% of LacI mNG foci, whereas 53BP1 is 43.2%. 36.9% of all VG foci colocalize with BRCA1 but not with 53BP1, whereas 53BP1 colocalizes without BRCA1 on only 7.4% of VGs. This suggests that DSB repair factors are recruited to most newly uncoated VGs, and that HDR machinery may be preferentially recruited over NHEJ. We have added this to the main text at lines 187-193.

2- Concluding that the viral genome is chromatinized based on the colocalization of gH2AX is a strong statement. Can the authors use their Lacl construct to pulldown viral DNA and validate nucleosome incorporation?

We thank the reviewer for this valuable suggestion. Suitable antibodies against Lacl and mNeonGreen are not commercially available, which makes this difficult. Instead, we have performed ChIP-qPCR of the VG by pulling down H2B to verify general nucleosome incorporation, and γ H2AX to verify incorporation of this variant (Figure 3C, and lines 179-182). The earliest timepoint at which we could detect γ H2AX signal compared with the IgG control was 12hpt, presumably because enough uncoating and chromatinization has occurred by then to be detected in a bulk assay like ChIP. This supports our suggestion of the superiority of single-cell assays like our imaging platform's ability to capture the earliest events.

3- The conclusion in figures 3 and 4 assumes that the foci detected using the Lacl construct are concatenated viral genome. The Lacl construct binds to viral genomes that have been converted to dsDNA. Can the authors rule out the possibility that the foci detected by microscopy are monomers of the viral genome? Does Lacl binding impact the efficiency of concatenation?

The experiments in Figure 5 using the split vector requires concatenation in order to obtain transgene expression, and therefore these data corroborate the conclusions in Figures 3 and 4 that HDR inhibition increases concatenation. Because the Figure 5 experiments are performed with "WT" U2OS cells with non-array VGs, and also show that HDR loss increases concatenation and expression, Lacl binding does not appear to impact concatenation. A limitation of the fixed cells and high-content imaging is indeed that we cannot rule out monomeric VGs. Therefore, to address the reviewer's comment, we have added live imaging of VG foci. When two VG foci contact each other each other, they either (1) touch each other transiently and split back into two foci, or (2) merge irreversibly into a single focus, which typically grows larger and brighter. When two foci irreversibly merge, we interpret this to suggest they are covalently linked/concatenated. We observe more of these events in B02 treated cells (Movie 1, 2, & 3) than DMSO treated cells (Movie 4, 5, & 6), which is a third line of evidence to corroborate the main findings. We have added this to the main text to at line 232.

4- In Fig 3, can the author comment on the nature of the foci observed in the DAPI channel?

We thank the reviewer for bringing this to our attention. The DAPI foci made sense to us since VGs are DNA and would therefore be expected to be DAPI stained. However, we only see these foci in images taken with the LSM900 – they do not appear in the high-content imaging performed on the Opera Phenix (which is a spinning disk confocal). Although the conclusions have not changed, to represent our findings better, we have replaced these images with primary images from the high-content experiments.

5- Important information is missing in the material and method (the method used to quantify the viral genome presented in fig 4I details on the information presented on the graphs – mean vs median
The constructs used for the CRISPR screen contain an mCherry expression cassette as well as a blasticidine and a GFP expression cassette that are dependent on concatenation. Have these elements been used at any point during the screen?

*We apologize for this oversight and have addressed the reviewer comment by including more details. Vector genome quantification methods have been added to the M&M section under the heading "**Vector genome quantification in transduced cells**" (Lines 477-489) and all missing instances of mean and/or SD have been clarified in all figure legends. Fluorescent and antibiotic resistance genes are encoded in the L+R pairs used in the screen, but low levels of leaky expression from these unsplit genes made them undesirable to assay concatenation with high sensitivity. Moreover, the most stringent measure of split transgene reconstitution is restoring enzymatic activity. We therefore only used LacZ for our screen and moving forward. Although the other elements were not used here, we included the full schematic of the L+R pair to accurately depict the reagents used in the study.*

How was the BRCA1 KO validated? Is this a clonal cell line?

We experienced moderate to severe growth defects in clonal BRCA1 KO lines, so we took an acute approach to knockout by nucleofecting BRCA1 guide plasmids carrying Blasticidin resistance into a Cas9 stable expressing line. After 72h of selection, cells were transduced with rAAVs, fixed at 48 hpt, immunostained for BRCA1, and imaged. BRCA1 staining was quantified as in Figure S3A and used to verify that BRCA1 expression is lost in significantly more cells than in scramble guide controls. We have added text to clarify this in the M&M starting at line 540.

Considering the limitations of this approach, we have added a more robust line of evidence that BRCA1 loss increases VG expression and foci formation via siRNA experiments. We have moved the original BRCA1 KO results to a supplemental figure (S2). Pooled siRNAs against BRCA1 (or a nontargeting control) were transfected into U2OS^{LacI-mNG cells}, and transduced with AAV2/2.LacO.mScarlet 24h later, when significant knockdown of BRCA1 protein is evident by western blot (Figure 4E). The increase in VG expression/foci formation observed in these experiments exceeds those by acute KO, likely because of the improvement in cell health with this method which allows more robust transduction and data collection. This confirms that BRCA1 negatively impacts transduction and LacI foci formation. We have summarized this in the main text (lines 216-228) and modified Figure 4 accordingly.

How was the inhibition of RAD51 confirmed?

One way that we confirmed B02 inhibition of Rad51 was to treat cells with B02 and Etoposide which showed a dramatic increase in γ H2AX staining (Figure S2E). Additionally, in figure 5H we show a loss of Rad51 colocalization upon B02 treatment. Throughout our study we observed similar experimental results when using B02 and Rad51 depletion by siRNA, and therefore we are confident that Rad51 is similarly inhibited in both treatments.

Reviewer #2 (Remarks to the Author):

In the study conducted by Maurer et al., it was observed that homologous recombination (HR) exerts an inhibitory effect on dual vector transduction. Through the implementation of a genome-wide gene KO screen, the authors illustrated that the depletion or inhibition of HR factors, such as BRCA1 and Rad51, leads to a significant enhancement in the reconstitution of a large split transgene. This improvement is attributed to an increase in both concatenation and expression from rAAV. While the manuscript employs innovative methods to unravel the biological processes of rAAV, it is important to note that certain pivotal experiments are necessary to rule out alternative hypotheses before definitive conclusions can be drawn.

Specific points:

In Fig. 2B, the authors demonstrate the widespread use of genome-wide CRISPR screening in various studies. While RAD51 and BRCA1 are mentioned as significant hits, it would enhance reader comprehension if the authors provided a comprehensive list of all enriched genes in Fig. 2B. Additionally, presenting this information in a table with fold changes and enrichment scores for each gene would be valuable.

We have now added Table 1, which lists all enriched genes and the following properties:

Pathway name	Gene ID	Gene Symbol	Average LFC	Average - log(p-values)	Number of perturbations
----------------	--------------------	--------------------	--------------------------------	--------------------------------

The reference to Table 1 is found on line 153 of the main text.

In Fig. 3B, the visualization reveals limited colocalization.

Although we do not expect 100% colocalization at any timepoint, to address the Reviewer's concern we have performed high-content imaging and analysis on cells 4hpt and found that 64.97% of the 1139 VG foci imaged colocalize with γ H2AX foci (line 178). We have added more representative images in Figure 3B to better support this finding.

In lines 182-186 and Fig. 4, prior to the conclusions drawn in lines 185-191, it is recommended to include Western blot validation of BRCA1 gene knockout efficiency. Given that BRCA1 is a DNA repair-associated protein and to be crucial for rAAV transduction, assessing whether BRCA1-KO affects cell cycle or DNA damage response, potentially influencing rAAV transcription, would be insightful.

We thank the reviewer for these suggestions which we have addressed, and this has strengthened our main conclusions. We have included a greater level of detail explaining the acute nature of the BRCA1 KO experiments (lines 205-211). Since the cellular material was limiting, we demonstrate a significant loss of BRCA1 by immunostaining (Figure S3A). Considering the limitations of these experiments, we tested siRNA KD and achieved more robust results. Therefore, we moved the acute CRISPR-Cas9 based KO approach to a supplementary figure (Figure S3), and replaced the main figure with siRNA knockdown of BRCA1 (Figure 4B-D), including western blots (Figure 4E).

HDR perturbations would indeed be expected to affect the DNA damage response, which is presented in our model and discussed in the text. To examine the effects on cell cycle, we have included flow cytometry analyses for all perturbations in Figure S4. BRCA1 siRNA does not significantly affect the proportion of cells in G1, S, or G2/M compared to the negative control siRNA (maximum of 3.5% difference, Figure S4 F&H), whereas RAD51 siRNA has a significant effect (maximum 9.9% difference, Figure S4 F&G). Comparing B02 to DMSO conditions, however, demonstrates pharmacological Rad51 inhibition does not recapitulate the effect on cell cycle of siRNA (Figure S4 A&B). Since we observe the strongest increases in VG foci formation under BRCA1 depletion and this perturbation has the smallest effect on cell cycle, it is unlikely that cell cycle differences induced by HDR loss are indirectly driving the observed transduction effects, although we have not completely ruled this out. We have added discussion of this to the manuscript accordingly (lines 249-256).

In lines 191-192, it would strengthen the manuscript to provide direct evidence, such as qPCR or NGS data, demonstrating the inhibition of concatenation and de novo episome formation.

We employed qPCR to provide an assessment of VG copy number as in Figure S3E, but we cannot use qPCR to assess the extent to which VGs are concatenated. We assume the reviewer is suggesting NGS to sequence ITR junctions, which could in theory quantify concatenation and ITR integrity. Unfortunately, it is well known in the AAV field that it is not possible to PCR through ITRs accurately with any standard methods, rendering NGS inapplicable. We have instead added live imaging of VG foci in treated cells, and observe more fusion events between individual foci in B02 treated cells (Movies 1-3) than DMSO treated cells (Movies 4-6).

In Fig. 4E-G, a Western blot is essential to validate the effect of Drug B02. Considering RAD51's significance in this study, it is advisable to include RAD51 knockout or knockdown experiments to further validate RAD51's function.

B02 inhibits Rad51 ATPase activity and is not known to affect protein levels, and we have included a western blot in Figure 4E to show that this is the case in our hands. Additionally, per the Reviewer's suggestion we have added siRNA for Rad51, and we demonstrate that we see the same effect as with B02. These additional experiments also allowed us to conclude that ATPase activity of Rad51 is important for the inhibitory effects on rAAV (Lines 234-235). We thank the reviewer for the valuable suggestion that strengthened the manuscript.

In Fig. 4J-K, clarification is needed regarding why colocalization was not observed.

We do not expect perfect colocalization of these foci, only some overlap as the mRNA hairpins are tethered at one end and should lie adjacent to the VG as diagrammed in Fig 4I. This overlap is easily detectable and quantifiable by the image analysis software (Figure 4K).

In Fig. 5, it is suggested to include Western blot analyses, cell viability assays, and cell cycle assessments to validate the inhibitory effects of B02 and NU7441. Furthermore, Fig. 5G should incorporate images to visually demonstrate the relevant findings.

LAFA fluorescence in Figure 5 graphs are normalized to a cell viability assay as described in the Materials and Methods section (lines 569-571). Since these drugs affect activity but not protein levels, we have included cell cycle analyses per

the reviewer's suggestion (Figure S3) for all perturbations. While we observe small effects on cell cycle in most conditions, they are usually too small to implicate cell cycle differences as the primary (and indirect) driver of the observed transduction effects. Nonetheless, we have not completely ruled this out and acknowledge it in the main text (lines 253 – 256).

Reviewer #3 (Remarks to the Author):

Summary

One of the critical drawbacks to the use of recombinant AAV gene therapy vectors is the limitation of the transgene size, precluding the possibility of using therapeutic transgenes that exceed the packaging capacity of the AAV capsid. This drawback can be overcome by using multiple rAAV vectors that are engineered to recombine in the host cell and reconstitute a large therapeutic transgene. However, these rearrangements have place at low efficiency. Little is known about how rAAV genomes recombine to form extrachromosomal concatemers and how they express. To address this deficiency in knowledge, in this manuscript, Maurer et al. have investigated the mechanisms of homologous recombination of rAAV vectors, dissecting how both concatemer formation and expression from rAAV vector genomes are related, to improve the efficiency of dual vector systems. Using a split dual vector system, the authors perform a CRISPR screen to identify the host factors mediating concatenation of rAAV genomes in the host nuclei. These assays identified the homologous recombination pathway of DNA repair as the mechanism facilitating rAAV rearrangements. Using a previously developed rAAV vector system containing LacO binding sites to label rAAV genomes, they determine how localization and distribution are impacted upon inhibition of the HR pathway. These discoveries are exciting for the field of gene therapy. While these findings are novel and of interest, the authors have made some conceptual and proposed mechanistic leaps that are not corroborated by experimental evidence. Claims regarding how these essential DNA repair proteins can be targeted by pharmacological intervention to improve rAAV transduction disregard detrimental effects to patients and are not necessary. These discrepancies should be addressed with corrections to their model or appropriate experimental evidence, which would further strengthen their conclusions. The specific suggestions for improvement are detailed below:

Major Comments

Line 113 – 115: The authors should provide more clarity on the difference between the dual vector systems used in the experiment. How is the reconstituted transgene efficiency affected by the additional AP segments in the Hybrid vectors? Do the trans-splicing (TS) vectors need validation from the Hybrid vectors? Were the cells infected with only one set of dual vectors at a time? These experiments and appropriate controls need to be described more fully.

The history of dual trans-splicing (TS) and hybrid (HB) vectors, including what is known and unknown mechanistically, is discussed in the introduction (lines 80-97), then referenced again in the Results section in line 116. In Figure 1E we demonstrate an increase in HB compared to TS split transgene reconstitution in our system, with flow cytometry that is consistent with the literature (and with the experiments in Figure 5). TS vectors do not need validation from HB vectors, as cells are only transduced with one set of dual vectors at a time. To clarify these points, we have revised the language and figures as follows in lines 124-131: “We transduced U2-OS cells with unsplit LacZ as a positive control, with either the TS or HB pair of dual vectors, or an empty capsid as a negative control, and used LAFA and flow cytometry to assay expression of LacZ. Flow cytometry of treated cells (Figure 1E) is consistent with previous observations by others: although HB dual vectors (purple) achieve higher transduction than TS (magenta) at a moderate dose of 1e4 total vg/cell (5e3 vg/cell of L vector plus 5e3 vg/cell of R vector), both the TS and HB dual vectors transduced at efficiencies two orders of magnitude lower than cells receiving the same dose of an unsplit LacZ vector (green).”

Figure 3B: An empty vector only control is needed for this experiment.

We have added a no vector control and the following to the main text at lines 169-171: “untransduced cells have no LacI-mNG foci (Figure S2A) and normal γ H2AX immunostaining patterns, with foci distributed throughout nuclei (Figure S2B).”

Why is the gH2AX timecourse assay not between 4- 48 hours as shown with previous experiments to show initial gH2AX recruitment and disassociation to initiate transcription? Because transcription cannot occur on packaged in gH2AX, the authors should provide some sort of explanation for these observations. RNA pol II should be utilized as a control.

We appreciate the reviewer's comment. We would like to clarify that in this experiment we are asking whether VGs are marked by γ H2AX, and when does this happen? We are not assessing transcriptional activation, and here the VG does not contain an expression cassette but instead possesses only a LacO array. We have modified the text to make this further clear. Analysis of kinetics of transcriptional activation and epigenetic marks on VGs over time is an exciting topic of future investigation, but outside the scope of this study.

Authors also need a positive control for DNA damage (such as using HU or UV radiation).

We have added an etoposide condition and see that there is a significant increase in γ H2AX signal (line 171 and Figure S2).

Line 165 – 166: The authors should provide appropriate references and experimental evidence to support their claim that the vector genomes are chromatinized immediately after uncoating and are subsequently flagged as DSBs with appropriate timecourses.

To support our claim that VGs are chromatinized and flagged as DSBs with additional experimental evidence, we have added ChIP-qPCR by both anti- γ H2AX and anti-Histone H2B at multiple timepoints (Figure 3C and lines 179-182). We believe this represents the first attempt to examine chromatinization of rAAV VGs at these early timepoints.

gH2AX is a marker for both host and viral DSB. Could this mean that rAAV is inducing double-stranded breaks on the host genome?

Since rAAV vectors neither package enzymes within the virion nor encode DNA modifying enzymes from the VG, there is no means to directly cause DSBs of the host genome, and therefore assume a mechanism for induction of DSBs. It is possible that rAAVs localize to host damage to capitalize on repair machinery that is active at these sites. This has been suggested to be the case for the AAV wt virus in Majumder et al (ref 32), and we have now expanded on these points in the discussion by adding lines 435-437.

Although the LacI-mNeon-Green is a great system to label rAAV vectors it does not indicate concatemerization to the same certainty at the Split donor and acceptor vectors. This drawback makes the LacI-mNeon-Green vector an inadequate system for measuring recombination.

Line 180: The authors should provide experimental evidence that increase in the size of the mNeonGreen foci are associated with concatenation.

We thank the reviewer for their appreciation of the split vector system, and we agree that imaging on fixed cells is not a direct measurement of concatenation. Therefore, to address the reviewer's comment, we have added live imaging of VG foci. When two VG foci contact each other each other, they either (1) touch each other transiently and split back into two foci, or (2) merge irreversibly into a single focus, which typically grows larger and brighter. When two foci irreversibly merge, we interpret this to suggest they are covalently linked/concatenated. We observe more of these events in B02 treated cells (Movie 1, 2, & 3) than DMSO treated cells (Movie 4, 5, & 6), which is a third line of evidence to corroborate the main findings. We have added this to the main text to at line 232.

Figure 4: The overall findings of Figure 4 require controls to show that the findings are not caused by derepression of the rAAV genome.

The Reviewer raises an intriguing point, and this will be an interesting topic of future investigation. We do not currently have a straightforward way of testing whether VGs start out repressed or primed and require derepression, versus activation, as the default state after double-strand conversion. In the absence of experimental suggestions, we have added the following text to the discussion section to acknowledge this limitation of the current study: "Another limitation

is that we cannot determine whether VGs enter and are repressed as the default state, which would require derepression and priming for activation. This is an interesting topic of future study.” (lines 437-439).

The difference between Fig. 4B and 4C are unclear. The authors should provide error bars and statistical significance in 4C.

We thank the Reviewer for highlighting this confusion. As labeled on the y-axes of these graphs, Figure S3B (formerly figure 4B) plots each cell by the number of LacI-mNG (VG) foci in the nucleus, and Figure S3C plots the percentage of cells that have any VG foci (> 0 foci) in scramble vs BRCA1 guide receiving cells. We have revised the legend to describe these graphs individually, and have added statistics to show significance per the reviewer’s comments.

These experiments should be accompanied by a BRCA1 western blot to confirm the knockout.

We have included a greater level of detail explaining the acute nature of the BRCA1 KO experiments (line 204 onward) and instead demonstrate a significant loss of BRCA1 by immunostaining (Figure S3A) because cellular material was limiting, making western blots challenging. The new BRCA1 siRNA experiments, shown in the revised figure 4B-D, include western blots in Figure 4E, and corroborate the acute BRCA1 KO studies.

Independent gRNAs are needed for the CRISPR knockout studies.

We employed pooled guides with multiple guide sequences, which is a common practice for knockout studies. Since the results are recapitulated by the newly added siRNA experiments (Figure 4), this validates that we are not looking at off-target CRISPR cutting effects.

The graph in Fig. 4I seems meaningless because it is not compared with any treatment conditions.

This graph is plotting the ratio of VG copy number in B02 treated cells to that of DMSO treated cells. We have revised the axes with more detailed descriptions and added a title to the graph to make this clearer.

Figure 5G: The authors should include representative images of VG foci with error bars and statistics. This experiment should also be done with DNA-PK inhibitor NU7441 to measure the impact of vector transduction/recombination upon NHEJ signaling.

We thank the reviewer for the valuable suggestion. Figure 5G is now representative images, and we have added the statistics to the quantification (now Figure 5H) per the reviewer’s requests. We attempted to perform high-content imaging analyses under Nu-7441 conditions, and unfortunately the drug treatment generated some auto-fluorescent cellular debris that prohibited reliable nuclear segmentation as well as LacI foci identification, selection, and quantification. More NHEJ mechanistic insight is of high interest and will be the topic of future investigation using amenable approaches.

Figure 6. This diagram shows gH2AX binding to the double-stranded rAAV genome. It has not previously been convincingly established by chromatin immunoprecipitation assays that gH2AX recognizes the rAAV vector as a double-strand break.

We have added ChIP experiments per the reviewer’s request and observe gH2AX occupancy by this method (Figure 3C). We thank the reviewer for this suggestion which has strengthened our main conclusions.

Minor Comments

Figure 1B-D: It is unclear if RNA processing proceeds or follows DNA recombination. How does this assay show more than two separate rAAV vectors can recombine? Control for rearrangement of the splice donors as transcripts (before actual recombination of the rAAV vectors).

We thank the reviewer for raising this point that we had not previously considered. To ensure that LacZ activity measured after dual vector transduction is from VG-VG recombination and not transcripts recombining, we transfected plasmids

containing the split vector genomes and unsplit controls (Figure S1 A-F), which do not have free ITRs and thus cannot concatenate in the way that capsid-delivered VGs do, but still actively express from their promoters. We do not observe LacZ reconstitution in this setting (Figure S1G), suggesting that concatenation must occur at the DNA level in order for a spliced transcript to reconstitute a split gene. We have added this to the main text (lines 134-139).

Figure 1E: This graph is uninformative and colors are confusing. What is the rearrangement frequency shown in this graph? Would it be possible to depict this in the form of a 3D plot for more clarity?

This is a standard flow cytometry histogram demonstrating that the split transgenes used in our study are orders of magnitude less efficient than their unsplit counterpart. The colors of the lines match the labels of the vector diagrams in panel C (which we have now enlarged and made bold), and the experimental approach is outlined in panel D. We are not sure what the reviewer means by “rearrangement frequency” since the word “rearrangement” does not appear anywhere in the manuscript. We have revised the Figure 1 legend to help clarify the plot in panel E.

Line 140: The authors refer to a pool performed to a generated list of candidate inhibitors for dual vector transduction. Please include hits from that screen. Authors need to clarify which genes they knocked out in this screen.

The genome-wide Brunello library is already described in lines 144-147. We have now added Table 1, which lists all enriched genes and the following properties:

Pathway name	Gene ID	Gene Symbol	Average LFC	Average - $\log(p\text{-values})$	Number of perturbations
--------------	---------	-------------	-------------	-----------------------------------	-------------------------

The table is cited in line 153 of the main text.

Line 142: Authors need to clarify the pathway analysis of significantly enriched genes taken from the negative or positive hits from the screen. If they are taken from the MUG negative it will indicate that recombination is inhibited by the CRISPR knockouts and the opposite for the MUG positive.

We have revised the language to clarify the reviewer’s points in both the main text (line 152-153), in the Figure 2 legend, and in the Materials and Methods section (lines 606-623). These changes also help clarify the following three reviewer comments. Please note that it is the MUG positive population, not the negative population, that would contain the inhibitors of concatenation.

Figure 2B: Representative dots of RAD51 and BRCA1 on the waterfall plot should be more prominent in color. What is the cutoff of the waterfall plot? What is the meaning of positive and negative population and how does it correlate to the negative and positive number demarcations?

We have changed the colors on the waterfall plot. There is no cutoff since we plotted every gene in the library by (2C) average Log-fold-change (LFC) and (2D) average $-\log_{10}(p\text{-value})$ (newly added) in the MUG positive population compared to the MUG-negative population, which are labeled on the y-axes of the graphs. We have also clarified the figure legend.

Figure 2C: This figure is uninformative and gene regulation categories need to make more sense. How is the “Cell cycle” category different from the “Cell cycle, Mitotic”? What is the overlap between these categories what genes would fit in one and not the other? What are some example genes that have been identified in the screen besides RAD51 and BRCA1 and how do they fit in each category? The authors should consider taking out this figure or focusing it only on the HR and NHEJ genes.

These categories are standardly used in gene ontology or pathway analyses. To address the reviewer’s concern, we have now included in Table 1 all the enriched genes in each pathway for our screen. For the full set of genes that belong to these categories, we direct the reader to Reactome.org. We think it is of value to the community to divulge all top pathways in our screen instead of just focusing on the ones we followed up on, and so we have retained this figure.

Line 183 Authors need a reference to the BRCA1 scrambled guides used in the U2-OS cells.

We have added the sequences to the Materials and Methods section (lines 546-550).

Line 191 – 192: The observation of Neon Green foci increase has been established to represent concatenation or episome formation (line 162) which seems to contradict the statement that “BRCA1 inhibits concatenation.” However, imaging showed a significant increase in the average number of VG (Neon Green) foci per cell. How does this data represent an inhibition of concatenation? A similar contradiction can be found in line 203 with RAD51 inhibition.

Since we propose that these cellular repair factors act as barriers to concatenation, we would expect that their inhibition would lead to an increase. We see an increase in foci under BRCA1 or Rad51 loss, which we interpret to mean that they are inhibitory to the concatenation process.

Line 237 – 238: Authors should not say this without disclosing how chemical inhibition of RAD51 or BRCA1 can affect the host cell and cause DNA damage and cancer.

We have added the following text to address the reviewer’s concern (lines 381-384) “Although prolonged use of DDR inhibitors is dangerous, our results suggest that a single, transient dose of drug preceding vector administration is sufficient to increase transduction, which may be safer to the patient than the toxic responses seen with extremely high vector doses.”

Line 241 – 251: In this section, although the author’s purpose seems to be to connect the dual vector as a model for concatenation to HR and HDR seems unnecessary and should be moved to the discussion.

We have included this section because it (1) introduces the fact that this heretofore untested mechanism somehow became dogma in the literature, and states we are the first to test it experimentally, and (2) explains the importance of including HB vectors in LAFA assays (because they are the original system used in the screen). We feel that these points therefore need to come before presenting the data, and therefore we have retained their position.

Figure 5A-5F: Is the dual vector system driving Homologous recombination or is driven by AAV recombination? In lines 251 -253, the authors sound like they are looking at the mechanisms of the split donor's system and not rAAV biology.

We would like to clarify that we believe the split donor vector genome is very much a part of the rAAV vector and its biology. Because it carries no machinery of its own, an rAAV is entirely dependent on host cell interactions for processing and expression of any transgene. We articulate the interdependence of host and vector biology in paragraph 2 of the introduction and in lines 60-67: “In its recombinant form, where a capsid of choice is packaged with an ITR-flanked transgene, rAAV VGs cannot replicate; but instead multiple VGs enter the nucleus and can concatenate into large episomal species¹¹. The absence of delivered enzymes implies that the post-uncoating nuclear steps of rAAV transduction are orchestrated by host cellular factors. Decades of observation have articulated the nuclear steps of transduction – uncoating, conversion to dsDNA, concatenation, expression – yet they remain observations, with little mechanistic detail and knowledge of facilitating host factors^{12,13}”. The dual vector system therefore reflects vector biology of rAAV.

Authors should include fluorescence threshold in all graphs. They should also explain (or speculate) why there is a decrease in fluorescence when more Vector dose is added.

We apologize for this oversight. The values in Figure 5 are background subtracted, so the threshold is zero. We have now clarified this in the figure legend. Presumably the Reviewer is referring to the Fluorescence of the unsplit vector at a moderate dose (i.e. Figure 5C, 1e5 vg/cell = 2000) exceeding that of a split vector at a high dose (i.e. Figure 5B, 1e6 vg/cell = 250). It is well known that split vectors are much less efficient than unsplit vectors, and we summarize the literature in the introduction before demonstrating it in our own hands in Figure 1E by flow cytometry. The data in Figure 5 are in good agreement with Figure 1C.

Line 261: The authors need to provide references to support their assertion that concatenation can be considered analogous to repairing chromosomal DSB.

We have revised the schematic and legend in Figure 6 to better illustrate this conclusion, have added a relevant reference, and made the following change to the main text: "It is not clear what VG structures are flagged as DSBs or other forms of damaged DNA, but possibilities include the blunt ends after ds-conversion⁴¹ (among other conformations, diagrammed in Figure 6)." at lines 326-329.

REVIEWER COMMENTS

Reviewer #1 (Remarks to the Author):

All the issues raised in the original manuscript have been addressed. It is noted that the authors put important effort into providing additional data to support their hypothesis, strengthening the revised manuscript.

The outcome of testing ATRi is quite intriguing and raises interesting questions about the potential role of this kinase in inhibiting transgene expression.

We thank Reviewer #1 for their support of our revised manuscript.

Reviewer #2 (Remarks to the Author):

The studies led by Maurer AC, et al. carried out a genome-wide gRNA knockout screen in U2-OS cells and identified the homologous recombination (HR) pathway plays a negative role to the genome recombination-based dual AAV vector transduction. They showed inhibition of HR-critical factor BRCA1 and Rad 51 significantly increased the transduction of AAV-dual vectors. In the revised version, while the authors address several technical questions, the study still has limitations regarding the understanding of AAV vector transduction biology and its applications.

Our study and results expand the understanding of rAAV transduction biology and its applications. We are looking forward to the deeper knowledge that will be built from this novel study.

Major points.

1. All the experiments were performed in U2OS cell lines. It is a human osteosarcoma cell line. The cells are highly proliferating and widely used for study of DNA damage and repair pathways. While both HR and NHEJ are available in U2OS cells, have the authors examined the function of the HR factor BRCA1 and Rad 51 in the transduction of the dual AAV vectors of other proliferating cell line cells?

U2-OS are ideal for rAAV studies since they are transduced at high levels using relatively low vector doses, and moreover have low background in LAFA assays. We have added an additional cell line derived from a very different tissue (placental epithelium), and see similar trends (new Figure S5, lines 305-307), albeit not as dramatic as in U2-OS, as BeWo have inherently higher assay background and lower transduction levels than U2-OS. These additional experiments address the concerns of Reviewer #2.

2. While HR generally prefers high fidelity repair during the S and G2 phases of the cell cycle of the proliferating cells, and NHEJ being used in other phases of the proliferating cells, and more importantly in non-proliferating cells/differentiated cells, when the HR pathway is limited or not available. Initially, the dual AAV vector-based trans-splicing was found and applied in muscle cells, where the recombination-based dual AAV vector transduction was high. Muscle cells are differentiated cells. In differentiated cells, where most of the AAV-based gene therapy applied, the HR machinery is not available. How do the authors address this reality?

The purpose of studying rAAV biology is to enable more versatile and reliable applications. Current clinical rAAV is limited to these nondividing tissues because they have the most predictable outcomes, and that is because most of the literature studies these easy to transduce tissues. If we want to broaden its use, it's critical to study rAAV in different model cell lines and systems, and with new tools and approaches such as those we introduce here.

There are several reports in the literature that AAV induces a DNA damage response, although there is still some debate in the field about this. If AAV induces DSB repair response, then the cell does not need to cycle for HDRi to be effective. We agree that testing the implications of our studies in additional cellular contexts is very important, but it is beyond the scope of the current study.

3. Please check if the NGS data of the lentiviral Brunello/Cas9 library screen have been deposited to the NCBI (Sequencing Read Archive (SRA)/BioProject numbers were not found in the manuscript).

We have deposited the raw reads and the accession numbers will be part of the published manuscript:

BioSample accessions: SAMN45223789, SAMN45223790

Temporary SubmissionID: SUB14916088

Release date: 2025-01-31, or with the release of linked data, whichever is first

Minor: Line 127 lacks references

We are grateful to the reviewer for catching that placeholder – we have resolved it.

Reviewer #3 (Remarks to the Author):

In this study, Maurer et. al. has exploited the natural concatenation property of rAAV vectors to generate large rAAV transgenes that are larger than the packaging capacity of the capsid. Leveraging this dual vector system, they have performed a CRISPR screen, identifying the homologous recombination (HR) pathway as the principal signaling pathway responsible for regulating rAAV concatenation and subsequent gene expression. Importantly, inhibition of HR proteins leads to increase in concatemer formation and expression of the split transgene, suggesting a transient dosage that might be applicable in-vivo for expression of the split transgenes in patients. Taken together, these studies identify a critical DNA repair pathway that is required for regulating the formation of rAAV concatemers in the host. These studies are sure to spur exciting discoveries in engineering gene therapy vectors that exceed the capacity of one capsid and also provide insights into how viral episomes persist in the nuclear environment. In this resubmission, the authors have gone to great lengths to provide detailed responses to all the reviewer's comments with extensive additional experiments, control studies and previously published data from the literature that validate their findings. These findings have significantly strengthened the overall conclusions of the study and are an important contribution to the field of vector biology. The overall conclusions and claims are supported by experimental evidence, the author's methodology is sound and meets the expected standards of the gene therapy and DNA damage fields. The authors have provided enough detail for the methods of the work to be reproduced.

We thank Reviewer #3 for their support of our revised manuscript.

REVIEWER COMMENTS

Reviewer #1 (Remarks to the Author):

All the issues raised in the original manuscript have been addressed. It is noted that the authors put important effort into providing additional data to support their hypothesis, strengthening the revised manuscript.

The outcome of testing ATRi is quite intriguing and raises interesting questions about the potential role of this kinase in inhibiting transgene expression.

We thank Reviewer #1 for their support of our revised manuscript.

Reviewer #2 (Remarks to the Author):

The studies led by Maurer AC, et al. carried out a genome-wide gRNA knockout screen in U2-OS cells and identified the homologous recombination (HR) pathway plays a negative role to the genome recombination-based dual AAV vector transduction. They showed inhibition of HR-critical factor BRCA1 and Rad 51 significantly increased the transduction of AAV-dual vectors. In the revised version, while the authors address several technical questions, the study still has limitations regarding the understanding of AAV vector transduction biology and its applications.

Our study and results expand the understanding of rAAV transduction biology and its applications. We are looking forward to the deeper knowledge that will be built from this novel study.

Major points.

1. All the experiments were performed in U2OS cell lines. It is a human osteosarcoma cell line. The cells are highly proliferating and widely used for study of DNA damage and repair pathways. While both HR and NHEJ are available in U2OS cells, have the authors examined the function of the HR factor BRCA1 and Rad 51 in the transduction of the dual AAV vectors of other proliferating cell line cells?

U2-OS are ideal for rAAV studies since they are transduced at high levels using relatively low vector doses, and moreover have low background in LAFA assays. We have added an additional cell line derived from a very different tissue (placental epithelium), and see similar trends (new Figure S5, lines 305-307), albeit not as dramatic as in U2-OS, as BeWo have inherently higher assay background and lower transduction levels than U2-OS. These additional experiments address the concerns of Reviewer #2.

2. While HR generally prefers high fidelity repair during the S and G2 phases of the cell cycle of the proliferating cells, and NHEJ being used in other phases of the proliferating cells, and more importantly in non-proliferating cells/differentiated cells, when the HR pathway is limited or not available. Initially, the dual AAV vector-based trans-splicing was found and applied in muscle cells, where the recombination-based dual AAV vector transduction was high. Muscle cells are differentiated cells. In differentiated cells, where most of the AAV-based gene therapy applied, the HR machinery is not available. How do the authors address this reality?

The purpose of studying rAAV biology is to enable more versatile and reliable applications. Current clinical rAAV is limited to these nondividing tissues because they have the most predictable outcomes, and that is because most of the literature studies these easy to transduce tissues. If we want to broaden its use, it's critical to study rAAV in different model cell lines and systems, and with new tools and approaches such as those we introduce here.

There are several reports in the literature that AAV induces a DNA damage response, although there is still some debate in the field about this. If AAV induces DSB repair response, then the cell does not need to cycle for HDRi to be effective. We agree that testing the implications of our studies in additional cellular contexts is very important, but it is beyond the scope of the current study.

3. Please check if the NGS data of the lentiviral Brunello/Cas9 library screen have been deposited to the NCBI (Sequencing Read Archive (SRA)/BioProject numbers were not found in the manuscript).

We have deposited the raw reads and the accession numbers are stated in the data availability statement (Lines 688-690):

BioSample accessions: SAMN45223789, SAMN45223790

Temporary SubmissionID: SUB14916088

Release date: 2025-01-31, or with the release of linked data, whichever is first

Minor: Line 127 lacks references

We are grateful to the reviewer for catching that placeholder – we have resolved it.

Reviewer #3 (Remarks to the Author):

In this study, Maurer et. al. has exploited the natural concatenation property of rAAV vectors to generate large rAAV transgenes that are larger than the packaging capacity of the capsid. Leveraging this dual vector system, they have performed a CRISPR screen, identifying the homologous recombination (HR) pathway as the principal signaling pathway responsible for regulating rAAV concatenation and subsequent gene expression. Importantly, inhibition of HR proteins leads to increase in concatemer formation and expression of the split transgene, suggesting a transient dosage that might be applicable in-vivo for expression of the split transgenes in patients. Taken together, these studies identify a critical DNA repair pathway that is required for regulating the formation of rAAV concatemers in the host. These studies are sure to spur exciting discoveries in engineering gene therapy vectors that exceed the capacity of one capsid and also provide insights into how viral episomes persist in the nuclear environment. In this resubmission, the authors have gone to great lengths to provide detailed responses to all the reviewer's comments with extensive additional experiments, control studies and previously published data from the literature that validate their findings. These findings have significantly strengthened the overall conclusions of the study and are an important contribution to the field of vector biology. The overall conclusions and claims are supported by experimental evidence, the author's methodology is sound and meets the expected standards of the gene therapy and DNA damage fields. The authors have provided enough detail for the methods of the work to be reproduced.

We thank Reviewer #3 for their support of our revised manuscript.